# Tree species and genetic diversity increase productivity via functional diversity and trophic feedbacks

Ting Tang[1,2†], Naili Zhang[3†], Franca J Bongers[1], Michael Staab[4], Andreas Schuldt[5], Felix Fornoff[6], Hong Lin[7], Jeannine Cavender-Bares[8], Andrew L Hipp[9], Shan Li[1], Yu Liang[1], Baocai Han[10], Alexandra-Maria Klein[11], Helge Bruelheide[12,13], Walter Durka[13,14], Bernhard Schmid[15]*, Keping Ma[1,2]*, Xiaojuan Liu[1]*

[1]State Key Laboratory of Vegetation and Environmental Change, Institute of Botany, Chinese Academy of Sciences, Beijing, China; [2]College of Life Sciences, University of Chinese Academy of Sciences, Beijing, China; [3]College of Forestry, Beijing Forestry University, Beijing, China; [4]Ecological Networks, Technical University Darmstadt, Darmstadt, Germany; [5]Forest Nature Conservation, Georg-August-University Göttingen, Göttingen, Germany; [6]Nature Conservation and Landscape Ecology, University of Freiburg, Freiburg, Germany; [7]Institute of Applied Ecology, School of Food Science, Nanjing Xiaozhuang University, Nanjing, China; [8]Department of Ecology, Evolution, and Behavior, University of Minnesota, St. Paul, United States; [9]The Morton Arboretum, Lisle, United States; [10]State Key Laboratory of Systematic and Evolutionary Botany, Institute of Botany, Chinese Academy of Sciences, Beijing, China; [11]Chair of Nature Conservation and Landscape Ecology, Faculty of Environment and Natural Resources, University of Freiburg, Freiburg, Germany; [12]Institute of Biology/Geobotany and Botanical Garden, Martin Luther University Halle-Wittenberg, Halle, Germany; [13]German Centre for Integrative Biodiversity Research (iDiv) Halle-Jena-Leipzig, Leipzig, Germany; [14]Department of Community Ecology, Helmholtz Centre for Environmental Research–UFZ, Halle, Germany; [15]Department of Geography, University of Zurich, Zurich, Switzerland

*For correspondence:
bernhard.schmid@geo.uzh.ch
(BS);
kpma@ibcas.ac.cn (KM);
liuxiaojuan06@ibcas.ac.cn (XL)

†These authors contributed equally to this work

**Abstract** Addressing global biodiversity loss requires an expanded focus on multiple dimensions of biodiversity. While most studies have focused on the consequences of plant interspecific diversity, our mechanistic understanding of how genetic diversity within plant species affects plant productivity remains limited. Here, we use a tree species × genetic diversity experiment to disentangle the effects of species diversity and genetic diversity on tree productivity, and how they are related to tree functional diversity and trophic feedbacks. We found that tree species diversity increased tree productivity via increased tree functional diversity, reduced soil fungal diversity, and marginally reduced herbivory. The effects of tree genetic diversity on productivity via functional diversity and soil fungal diversity were negative in monocultures but positive in the mixture of the four tree species tested. Given the complexity of interactions between species and genetic diversity, tree functional diversity and trophic feedbacks on productivity, we suggest that both tree species and genetic diversity should be considered in afforestation.

## Editor's evaluation

This study uses a landmark experiment to provide compelling evidence that two mechanisms (increased trait space and biological interaction through herbivores and soil fungi) interact with

intra- and interspecific genetic diversity to promote forest productivity. These results will be important to foresters and molecular ecologists looking to improve their practices to increase or maintain forest ecosystem functions.

## Introduction

Biodiversity is essential for maintaining ecosystem functioning and nature's contributions to people (*Cardinale et al., 2012*; *Diaz et al., 2019*). Ongoing biodiversity loss has received widespread concern from the international community (*Ceballos et al., 2015*). Expanding our research focus to multiple dimensions of biodiversity helps us to better predict the consequences of biodiversity loss and prioritize the different dimensions of biodiversity in conservation efforts (*Cardinale et al., 2012*). Whereas many studies related to biodiversity–ecosystem functioning (BEF) have focused on how inter-specific diversity (e.g., the number of species) affects key ecosystem functions such as plant produc-tivity (*Hector et al., 1999*; *Huang et al., 2018*; *Tilman et al., 2001*), relatively few have addressed the effects of intraspecific diversity (such as genetic variation within a species). Furthermore, the effects of intraspecific diversity show an inconsistent picture: genetic diversity has promoted plant commu-nity productivity in herbaceous plant communities (*Crutsinger et al., 2006*; *Kotowska et al., 2010*) but not in forests (*Bongers et al., 2020*; *Fischer et al., 2017*). To get a better understanding of how genetic diversity influences plant productivity in forests and thereby help guiding afforestation priori-ties, we need to disentangle the underlying mechanisms.

Functional trait diversity, in short functional diversity, is expected to promote community produc-tivity because different species or genotypes with diverse traits may use resources in complementary ways and then enhance the total utilization of resources in the whole community (*Diaz and Cabido, 2001*; *Figure 1a*). Thus, functional diversity, mostly quantified as the variation of species functional trait means in a plant community, has been used to explain how plant species diversity impacts plant productivity (*Cadotte et al., 2011*; *Diaz et al., 2007*; *Hillebrand and Matthiessen, 2009*). Although genetic diversity has been shown to cause substantial trait variation within species (*Bongers et al., 2020*), and intraspecific trait variation may have strong effects on plant productivity (*Des Roches et al., 2018*; *Koricheva et al., 2018*), the extent to which genetic diversity can influence tree produc-tivity through increased functional diversity is still unclear.

Trophic feedbacks, which result from the interactions of plants of different species or genotypes with other trophic groups, have been suggested as an additional mechanism underpinning positive biodiversity effects (*Laforest-Lapointe et al., 2017*). Trophic feedbacks can enhance the performance of species or genotype mixtures either by reducing herbivore damage through enhancing the diversity of nutrient traits (*Wetzel et al., 2016*) and chemical traits (*Bustos-Segura et al., 2017*) or enhancing diversity of beneficial mutualists (e.g., mycorrhizal fungi; *Semchenko et al., 2018*; *Figure 1b*). These trophic feedbacks can be affected by plant functional diversity (*Schuldt et al., 2019*) and other factors (e.g., structural diversity; *Schuldt et al., 2019*), which may provide more niche opportunities for other trophic groups. However, whereas many studies have analyzed how plant diversity influences other trophic groups (*Scherber et al., 2010*; *Schuldt et al., 2019*) or how trophic interactions affect plant performance (*Eisenhauer, 2012*; *Semchenko et al., 2018*), the effects of plant diversity on other trophic groups and the feedbacks of these on productivity have rarely been analyzed in combination.

In real-world ecosystems, plant species diversity and genetic diversity can hardly be expected to influence ecosystems separately (*Vellend and Geber, 2005*). Previous studies of herbaceous plant communities have shown that the intensity of competition among species can be lowered by increased genetic diversity, which modifies the relationship between plant species diversity and plant produc-tivity (*Schöb et al., 2015*). Likewise, the relative extent of plant intraspecific variation in functional traits, partly due to genetic diversity, has been shown to decrease with the increase in species diversity (*Siefert et al., 2015*). Although there are few forest experimental studies in which species and genetic diversity are simultaneously manipulated, most of them only compared their relative importance on ecosystem functions (*Abdala-Roberts et al., 2015*; *Koricheva et al., 2018*), and we barely know their interactive effects via functional diversity and trophic feedbacks on plant productivity.

Here, we disentangle how tree species diversity and genetic diversity affect tree community productivity via the impact of tree functional diversity and trophic feedbacks. We use data from a long-term tree species × genetic diversity experiment in a subtropical forest (*Bruelheide et al., 2014*;

**eLife digest** Biodiversity, the richness of species in a given ecosystem, is essential for maintaining ecological functions. This is supported by many long-term biodiversity experiments where researchers manipulated the numbers of tree species they planted in a forest and then evaluated both its productivity (how much biological material the forest produced in a given timeframe) and the health of its trees. This work contributed to our understanding of forest ecology and paved the way for better reforestation approaches. The most important observation was that diverse forests, which contain several tree species, are more productive and healthier than monocultures where a single tree species dominates. However, it remained unclear what the role of genetic diversity within individual tree species is in determining productivity and health of forests.

Tang, Zhang et al. set out to improve on previous studies on tree genetic diversity and community productivity by looking at two possible mechanisms that might affect the productivity of a forest ecosystem using publicly available data. First, they looked at the diversity of traits found within a tree population, which determines what resources in the ecosystem the trees can exploit; for example, trees with varied specific leaf areas (that is the ratio between a leaf's area and its dry mass) have more access to different intensities of sunlight for photosynthesis, allowing the whole forest to gain more biomass. Second, they considered interactions with other organisms such as herbivore animals and soil fungi that affect tree growth by either consuming their leaves or competing for the same resources.

Tang, Zhang et al. used a mathematical model to interpret a complex dataset that includes multiple parameters for each of four types of forest: a forest with a single tree species seeded from a single parent tree (which will have low species and genetic diversity), a forest with a single tree species seeded from several parent trees (low species diversity and high genetic diversity, due to the diversity of parents), a forest with four tree species each seeded from a single parent tree (high species diversity and low genetic diversity), and a forest with four tree species each seeded from several parent trees (high species and genetic diversity).

Using their model, Tang, Zhang et al. determined that species diversity promotes productivity because the increased diversity of traits allows trees to exploit more of the surrounding resources. Genetic diversity, on the other hand, did not seem to have a direct effect on overall productivity. However, greater genetic diversity did coincide with an increase in the diversity of traits in forests with a single tree species, which led to a decrease in damage to tree leaves by herbivores. This suggests that high genetic diversity in species-rich forests is likely also beneficial as herbivores are less able to damage tree foliage. As expected, in single-species forests with both low and high genetic diversity, higher soil fungi diversity was associated with a loss in productivity. Interestingly, in forests that had high species and genetic diversity, this effect was reversed, and higher genetic diversity reduced the loss of productivity caused by soil fungi, resulting in higher productivity overall.

These results should be considered in reforestation projects to promote genetic diversity of trees on top of species diversity when replanting. How genetic diversity leads to downstream mechanisms that benefit community productivity is not fully understood and future research could look at what specific genetic features matter most to help select the ideal mixture of trees to maximize productivity and increase the land's ecological and economic value.

Biodiversity–Ecosystem Functioning Experiment China Platform [BEF-China], https://www.bef-china.com). Tree species diversity (one or four species per plot) and genetic diversity (one or four seed families per species per plot) were manipulated in a factorial design to generate four plant diversity levels (*Figure 1c*). We measured five morphological and chemical leaf traits, which have been shown to relate to resource acquisition (*Cornelissen et al., 2003*) and can have substantial variation both among and within species (*Albert et al., 2010*). Functional diversity was calculated as the variation of these five traits among seed families (*Laliberté and Legendre, 2010*). We quantified trophic interactions either by direct measurements of interactions (i.e., herbivory) or using the diversity of the trophic group (i.e., soil fungi) as a proxy to capture unspecific interactions potentially underpinning BEF relationships (*Delgado-Baquerizo et al., 2016*). Specifically, we tested whether tree species and genetic diversity increased tree community productivity via increased functional diversity (*Figure 1a*)

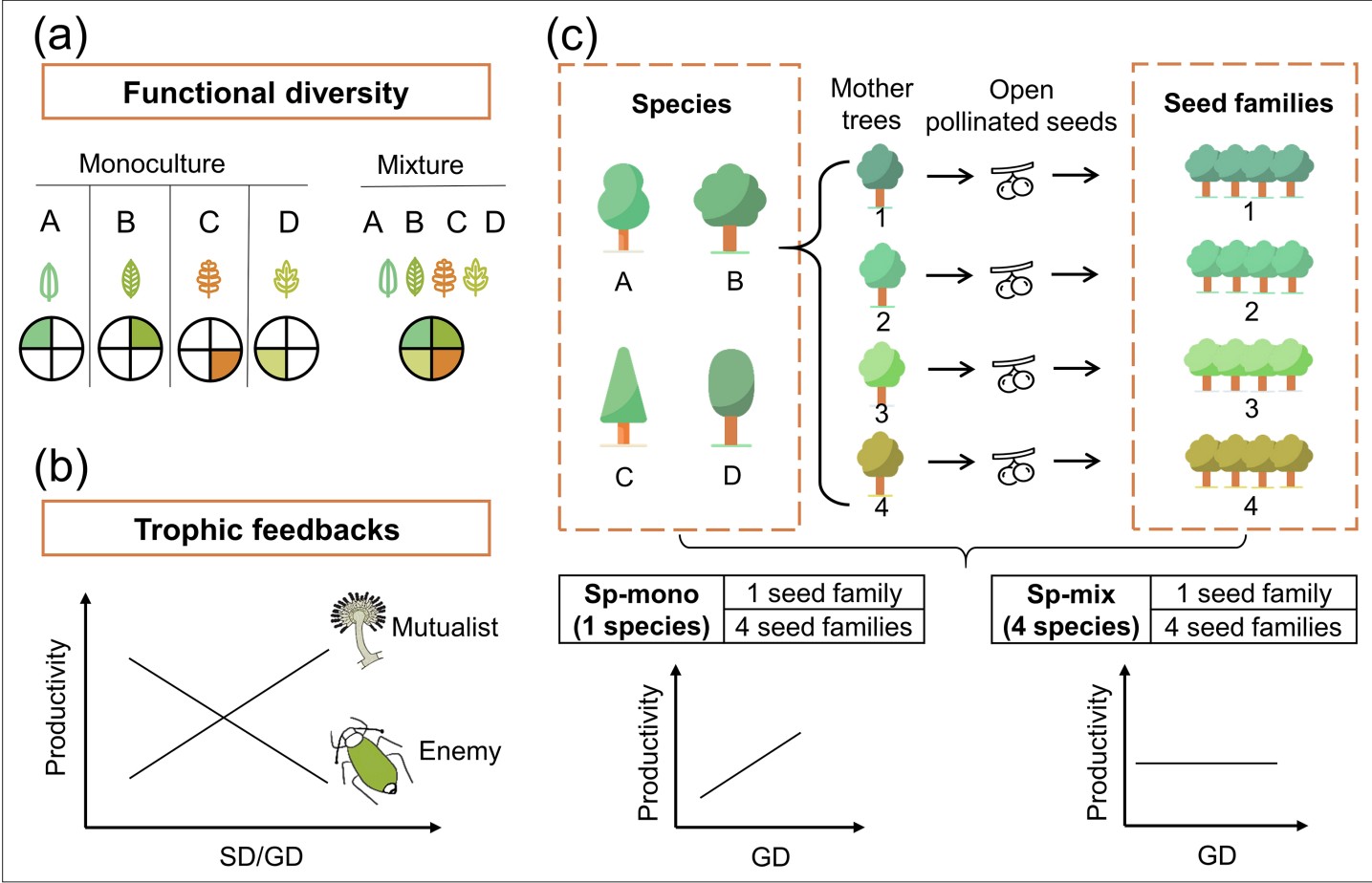

**Figure 1.** Conceptual illustration of the effects of functional diversity (**a**) and trophic feedbacks on tree productivity (**b**) and the species × genetic diversity experimental design (**c**). (**a**) shows resources for plant growth or other trophic groups in complementary ways due to functional diversity: the four hypothetical species/genotypes (A, B, C, D) with different functional traits (indicated by colored leaves) are able to use a heterogeneous resource (indicated by colored segments), thereby resulting in increased plant growth or providing niche opportunities for other trophic groups (*Diaz and Cabido, 2001*). (**b**) shows the mechanism of trophic feedbacks: with the increase in species diversity (SD) or genetic diversity (GD), negative feedbacks of enemies (e.g., herbivores) on tree productivity decrease due to diluted densities (*Duffy, 2003*) and positive feedbacks of mutualists on tree productivity increase due to increased diversity (e.g., mycorrhizal fungi; *Semchenko et al., 2018*). (**c**) We represent tree species and genetic diversity by the number of species and seed families (all seeds from the same mother tree are defined as a single seed family), respectively. Species diversity and genetic diversity per plot were both 1 or 4, resulting in a full factorial design of species × genetic diversity. We hypothesize that the positive effects of tree genetic diversity should be stronger in tree species monocultures (Sp-mono) than mixtures (Sp-mix).

and trophic feedbacks (*Figure 1b*). Furthermore, we tested whether the effects of genetic diversity were more important in species monocultures than in species mixtures because in the latter case genetic diversity between species may compensate for genetic diversity within species (*Figure 1c*).

## Results

### Direct bivariate relationships between tree diversity, trophic interactions, and tree community productivity

Using linear mixed-model analyses, we tested the effects of species diversity and genetic diversity within species on trophic interactions and community productivity. Overall, tree community productivity was significantly higher in the four-species mixture than in the four-species monocultures (*Figure 2a*), while genetic richness had no main effect on tree productivity in the bivariate analyses (*Figure 2a*). Tree functional diversity was higher in the species mixture than in the species monocultures and was also higher in genetic mixtures than genetic monocultures (*Figure 2b*). The effects of genetic diversity on tree functional diversity, herbivore leaf damage, and soil fungal diversity differed

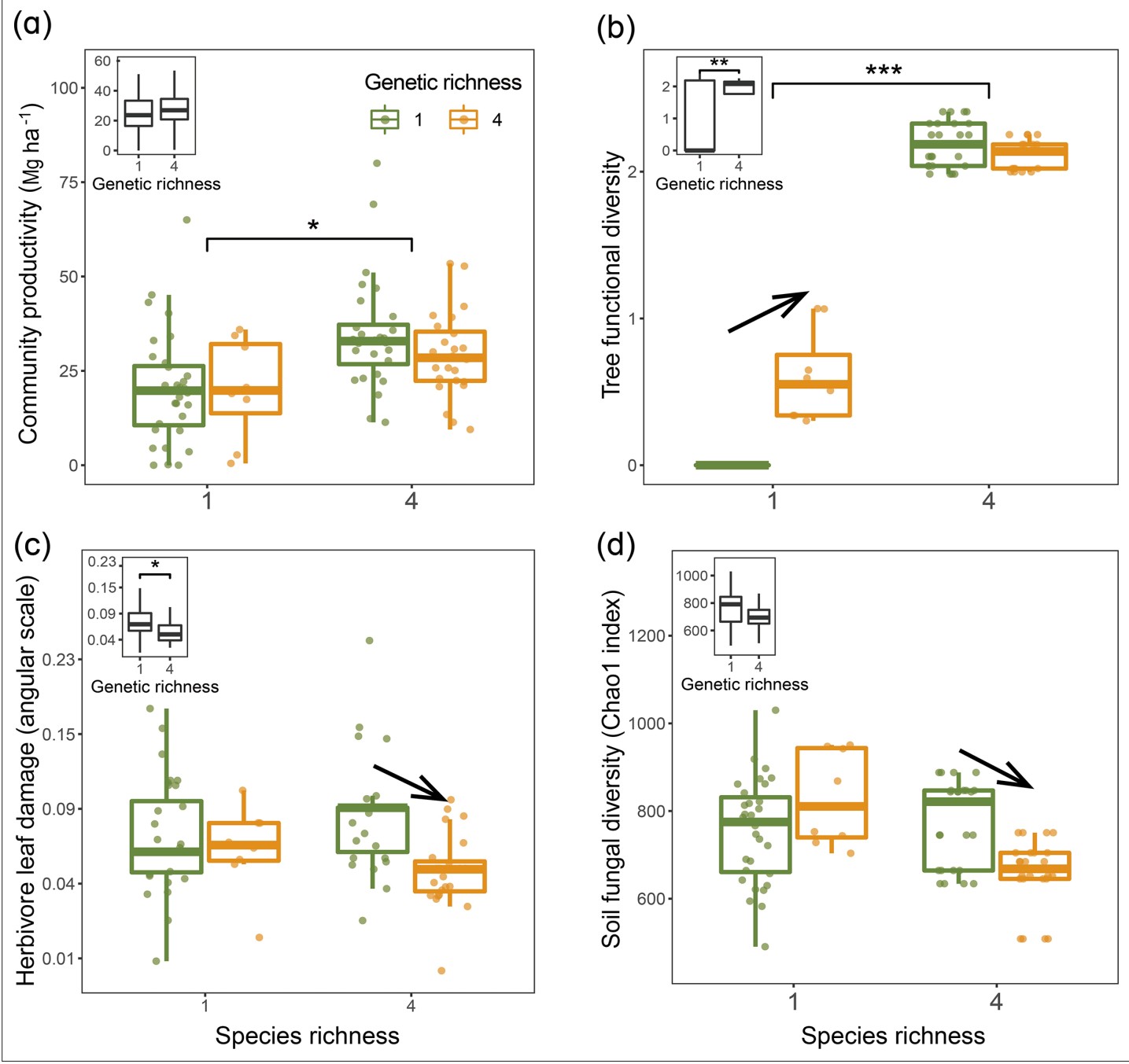

**Figure 2.** Tree community productivity, tree functional diversity, and trophic interactions in tree communities of low vs. high species and genetic richness. The following effects were tested in linear mixed-effects models (LMMs) (n=92): species richness main effect (left vs. right pair of bars in each panel), genetic richness main effect (inset on upper left in each panel), genetic richness effect within each species richness level (arrows between bars within pairs). (a) tree community productivity, (b) tree functional diversity, (c) herbivore leaf damage, and (d) soil fungal diversity. The lower and upper hinges of the bars correspond to the first and third quartiles (the 25th and 75th percentiles); the lower and upper whisker extends from the hinge correspond to 1.5 * interquartile range (third quartiles - first quartiles). Asterisks indicate statistical significance (*** $p<0.0001$, ** $p<0.001$, * $p<0.05$); solid arrow indicates ($p<0.05$, without arrow indicates $p>0.1$). Details of the fitted models are given in *Appendix 2—table 1*.

The online version of this article includes the following figure supplement(s) for figure 2:

**Figure supplement 1.** Effects of tree species diversity and genetic diversity on tree functional diversity calculated from traits measured on individual trees.

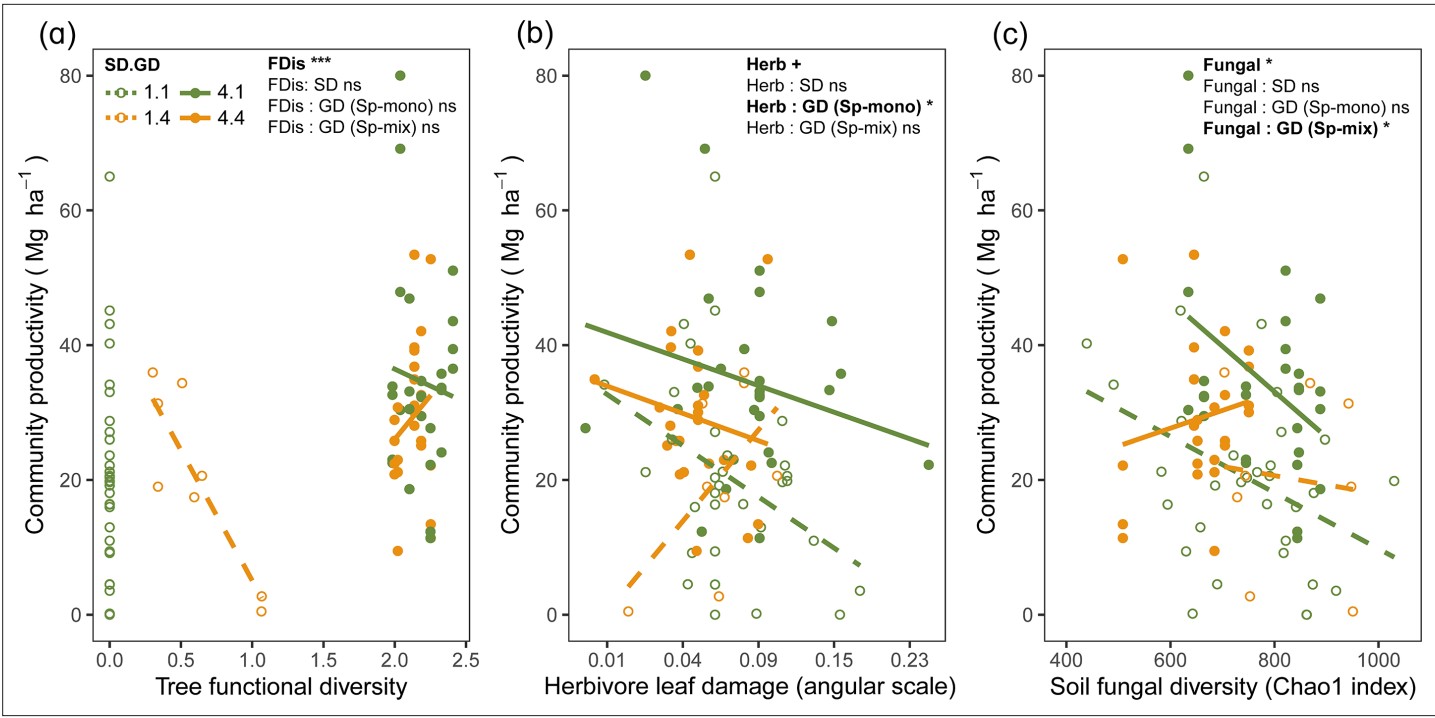

**Figure 3.** Bivariate relationships between tree community productivity and tree functional diversity (**a**), herbivory (**b**), and soil fungal diversity (**c**). Green unfilled/dashed symbols represent genetic monocultures in species monocultures, green filled/solid symbols represent genetic monocultures in species mixture, orange unfilled/dashed symbols represent genetic mixtures in species monocultures, orange filled/solid symbols represent genetic mixture in species mixture. FDis, tree functional diversity; Herb, herbivore damage; Fungal, soil fungal diversity; Sp-mono, species monocultures; Sp-mix, species mixtures; SD, species diversity; GD, genetic diversity. ':' indicates the interaction effects. Asterisks indicate statistical significance (*** p < 0.0001, ** p < 0.001, * p < 0.05, + p < 0.1, and ns p > 0.1).

The online version of this article includes the following figure supplement(s) for figure 3:

**Figure supplement 1.** Effects of tree functional diversity calculated from traits measured on individual trees on community productivity.

between species monocultures and species mixtures. Tree functional diversity in four seed-family species monocultures was larger than in one seed-family species monocultures but did not differ between species mixtures with four or one seed family per species (*Figure 2b*). However, when we calculated functional diversity based on measurements taken on individual trees rather than based on seed-family means, only species diversity but not genetic diversity had effects on tree functional diversity (*Figure 2—figure supplement 1*), indicating additional within-seed-family variation masking some of the between-seed-family variation. Furthermore, both herbivore leaf damage and soil fungal diversity were similar in one and four seed-family species monocultures but lower in species mixtures with four than species mixtures with one seed family per species (*Figure 2c and d*). Due to the equal representation of seed families across tree diversity treatments (*Appendix 1—table 1*), we did not find any significant effects of tree species and genetic diversity effects on community-weighted means (CWMs) of tree functional traits (*Appendix 2—table 1*).

Tree functional diversity calculated using either seed-family means or individual tree values had positive overall effects on community productivity, but this effect was mainly due to an increase in functional diversity from species monocultures to mixtures (*Figure 3a*, *Figure 3—figure supplement 1*). Herbivore leaf damage and soil fungal diversity showed negative overall effects on tree productivity (marginally significant for herbivory and significant for fungal diversity; *Figure 3b and c*). Furthermore, the effects of herbivore damage were different between genetic monocultures and genetic mixtures in species monocultures (*Figure 3b*), while the effects of soil fungal diversity were different between genetic monocultures and genetic mixtures in the species mixture (*Figure 3c*).

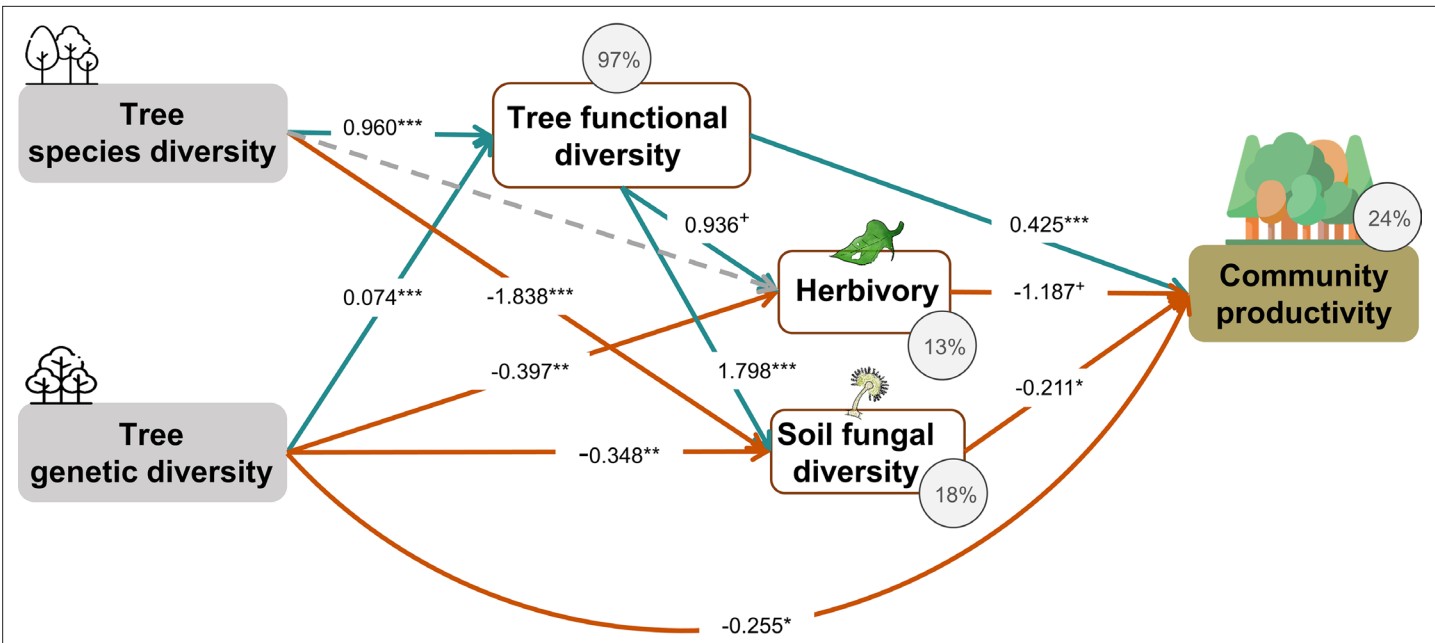

**Figure 4.** Effects of tree diversity on higher trophic levels and tree community productivity (global Fisher's C = 1.677, DF = 4, p = 0.795). Positive and negative paths are indicated in green and orange, respectively. The standardized path coefficients are indicated by the numbers, statistical significance is indicated by asterisks (*** p < 0.0001, ** p< 0.001, * p < 0.05, and + p < 0.1), and the explained variance of dependent variables is indicated by the percentage values. The gray dashed line indicates a nonsignificant (p > 0.1) pathway in the final model. The direct effect of tree species diversity on tree community productivity was removed in the model because it was not significant (p > 0.5) and the removal reduced the AICc by more than 2 (ΔAICc = 3.269).

The online version of this article includes the following figure supplement(s) for figure 4:

**Figure supplement 1.** Effects of tree diversity on higher trophic levels and tree community productivity with functional diversity calculated using trait values of individual trees (global Fisher's C = 119.558 DF = 4, p=0.001).

**Figure supplement 2.** Initial structural equation model (SEM) used in this study.

## Functional diversity and trophic feedbacks explain the effects of tree species and genetic diversity on tree productivity

Tree species and genetic diversity promoted tree community productivity as well as trophic interactions primarily indirectly through functional diversity (*Figure 4*). The increase in functional diversity was larger for increasing species diversity than for increasing genetic diversity (standardized path coefficient = 0.960 vs. 0.074, *Figure 4*). Herbivory and soil fungal diversity reduced tree community productivity (*Figure 4*, see also *Figure 3b and c*). Overall, tree diversity had contrasting effects on tree community productivity through different mechanisms: species and genetic diversity promoted tree functional diversity, which increased productivity directly but reduced it indirectly via negative feedbacks of herbivory and soil fungal diversity. However, species and genetic diversity also had positive indirect effects on community productivity via reduced soil fungal diversity (and genetic diversity additionally via reduced herbivory; *Figure 4*). Whereas tree functional diversity and trophic feedbacks explained all effects of tree species diversity on productivity, there remained a direct negative effect of tree genetic diversity on productivity, which could not be explained by the measured covariates (*Figure 4*). The analysis that functional diversity calculated from measurements on individual trees also showed that tree species diversity and genetic diversity affect community productivity via tree functional diversity and trophic feedbacks, although the effects of functional diversity were less pronounced (*Figure 4—figure supplement 1*), possibly because functional diversity calculated from individual trees included more response functional diversity (*Sapijanskas et al., 2014*) than did functional diversity calculated form seed-family means. Additionally, removing the path between genetic diversity and functional diversity did not change the remaining results we found by using functional diversity calculated from seed-family means (*Figure 4*, *Appendix 3—figure 1*).

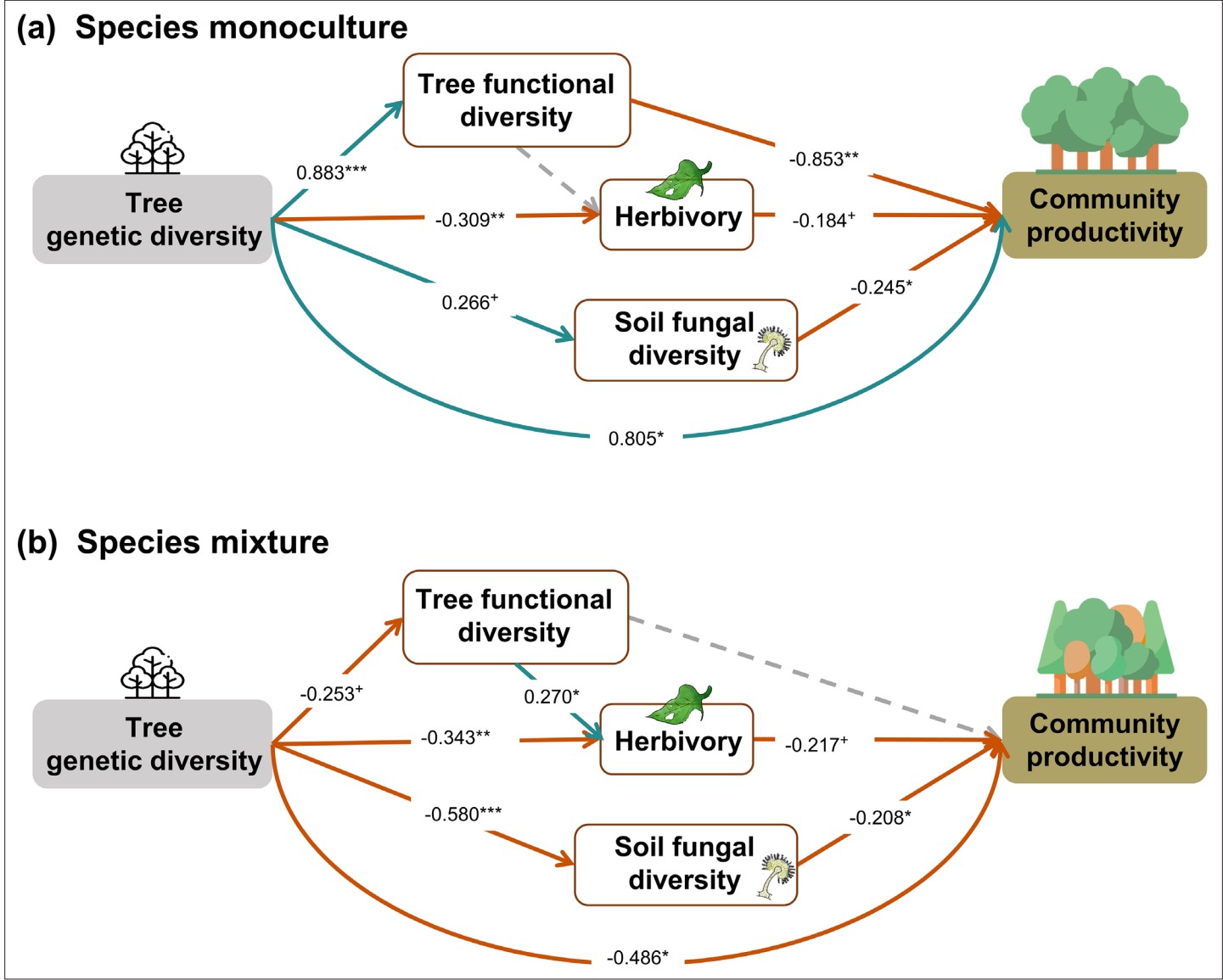

**Figure 5.** Effects of tree genetic diversity on higher trophic levels and tree community productivity in tree species monocultures (**a**) and the mixture of the four tree species (**b**). The results were obtained by a multigroup structural equation models (SEM) (global Fisher's C = 3.416, DF = 4, p = 0.491). Positive and negative paths are indicated in green and orange, respectively. The standardized path coefficients are indicated by the numbers, and statistical significance is indicated by asterisks (*** $p < 0.0001$, ** $p < 0.001$, * $p < 0.05$, and + $p < 0.1$). Gray dashed lines indicate nonsignificant (p > 0.1) pathways in the final model. The nonsignificant path from tree functional diversity to soil fungal diversity was removed because the removal decreased the AICc by more than 2 (ΔAICc = 2.176). Multigroup SEM analyses first test the interaction (explanatory variable × groups) in the whole model using the full dataset and then estimate the local coefficient for each path by using different datasets (the full dataset or group sub-datasets [species richness = 1 or 4, respectively]) depending on the significance of explanatory variable × groups interactions. Thus, we could not get the percentage of the explained variance in the local multi-group SEM model. All the paths were allowed to be different between species monocultures and mixtures (none of the paths was constrained manually beforehand); the interaction statistics of the multigroup model, and the explained variance of the whole model for each response is shown in *Appendix 2—table 5*.

The online version of this article includes the following figure supplement(s) for figure 5:

**Figure supplement 1.** Effects of tree genetic diversity on higher trophic levels and tree community productivity in tree species monocultures (**a**) and mixtures of four tree species (**b**) with functional diversity calculated using trait values of individual trees.

**Figure supplement 2.** Initial structural equation models (SEM) structure of genetic diversity effects in both species monocultures and mixtures.

## Effects of tree genetic diversity in species monocultures and species mixtures

When the above analysis was split into two (*Figure 5*), in contrast to our hypothesis, we found that tree genetic diversity negatively affected community productivity via functional diversity and soil fungal diversity in species monocultures and had positive effects via soil fungal diversity in the species mixture (see also *Figures 2b and 3a*). The results obtained with functional traits calculated from measurements on individual trees showed weaker effects of genetic diversity on functional diversity (path coefficient = 0.193 vs. 0.883) but did not change the significance and direction of the effects of genetic diversity on productivity via functional diversity in species monocultures. Additionally, the effects of functional diversity on tree productivity in species mixtures were positive when using functional diversity calculated from measurements on individual trees but were nonsignificant when using functional diversity calculated from seed-family means (*Figure 5*, *Figure 5—figure supplement 1*). Positive indirect effects through herbivory (resulting from two negative paths from genetic diversity to herbivory and form herbivory to community productivity) were similar in both species monocultures and mixtures. Using functional diversity calculated from measurements on individual trees did not change the effects of genetic diversity via trophic feedbacks, except that the effects of herbivory on productivity became nonsignificant from marginally significant. When we excluded the effects of genetic diversity on functional diversity in the analyses using functional diversity calculated from seed-family means, the remaining path coefficients did not change (*Figure 5*, *Appendix 3—figure 2*). The negative indirect effect of genetic diversity on community productivity via functional diversity in species monocultures, which contrasts with the combined analysis, was counterbalanced by a positive direct effect of genetic diversity on productivity, indicating that other aspects than those included with the five functional traits measured were important.

## Discussion

Our study demonstrates that manipulating tree species and genetic diversity in a factorial design can reveal effects of both as well as their interaction on measured ecosystem variables. Regarding our first hypothesis, we found that tree species diversity and genetic diversity can increase tree community productivity via increased functional diversity and trophic feedbacks as predicted. This suggests complementary resource-use and biotic niches, respectively, as mechanisms underpinning the biodiversity effects (*Turnbull et al., 2016*). Nevertheless, compared with the effects of species diversity, the effects of genetic diversity on tree community productivity through functional diversity were weaker, whereas the effects of genetic diversity on trophic interactions were strong (see *Figure 4*, *Figure 4—figure supplement 1*), indicating that the mechanisms underpinning the effects of genetic diversity may in part differ from those underpinning the effects of species diversity, as we will discuss below. Regarding our second hypothesis, we found that the effects of tree genetic diversity on productivity via functional diversity and soil fungal diversity were negative in tree species monocultures but positive in the species mixture, which differed from our predictions. In the following, we discuss these results in more detail.

### Tree species and genetic diversity drive tree community productivity mainly via functional diversity and trophic feedbacks

Although only species diversity but not genetic diversity was found to affect tree productivity in binary analyses, both kinds of diversity positively affected tree community productivity and trophic interactions via functional diversity according to our structural equation models (SEMs) depicted in the corresponding path-analysis diagrams (see *Figure 4*). Tree functional diversity appeared to enhance complementary resource acquisition at community level (*Kahmen et al., 2006*; *Marquard et al., 2009*; *Williams et al., 2017*), which consequently enhanced tree community productivity. Meanwhile, tree functional diversity also provided more niche opportunities to benefit generalist herbivores and soil fungi, which reduced tree community productivity, as has been found for these tree species in a parallel field study nearby (*Brezzi et al., 2017*). It is expected that herbivory has negative effects on plant productivity via the reduction of leaf area (*Zvereva et al., 2012*) and photosynthesis of remaining leaves (*Nabity et al., 2009*), and via trade-offs between growth and herbivore defense (*Züst and Agrawal, 2017*). The negative effects of soil fungal diversity on productivity correspond

with the finding that the majority of these fungi were saprophytes (*Appendix 2—figure 1*), competing with plants for resources (*Kaye and Hart, 1997*; *van der Heijden et al., 2008*). Indeed, in a related study in the same region, the diversity of saprophytic fungi had been found to decrease ecosystem multifunctionality (*Schuldt et al., 2018*).

Indirect positive effects of species and genetic diversity – remaining after accounting for paths via functional diversity – via reduced herbivory and soil fungal diversity further increased community productivity (see *Figure 4*). This finding corresponds to previous findings that plant diversity may reduce negative feedbacks of other trophic groups by decreasing the density and diversity of specialist enemies (e.g., *Duffy, 2003*; *Jactel and Brockerhoff, 2007*).

To account for possible effects of functional diversity within seed families, we also calculated functional diversity based on measurements of individual trees (see 'Materials and methods'). Overall, the results from this novel method still support our hypotheses that tree species diversity and genetic diversity affect community productivity via tree functional diversity and multi-trophic feedbacks (*Figure 4—figure supplement 1*), although compared with the typically used 'mean' method, the novel method includes more variation among individuals, which partly reflects responses of traits to the particular local environment (*Sapijanskas et al., 2014*); and this may have blurred the mean effects of tree genetic diversity and species diversity (*Figure 4*, *Figure 4—figure supplement 1*). At the same time, the results indicate that the seed-family means method may bring an artifact to the effect of genetic diversity on functional diversity because of the zero value of functional diversity in genetic monocultures of single species (1.1 communities). However, excluding the path between genetic diversity and functional diversity did not affect remaining paths, indicating that the partly artificial relationship between genetic diversity and functional diversity did not distort the path model in general (*Figure 4*, *Appendix 3—figure 1*).

Even after accounting for tree functional diversity and trophic feedbacks, we still detected a direct negative effect of tree genetic diversity on tree productivity, while the direct effect of tree species diversity was fully explained by functional diversity and trophic feedbacks. This suggests that aspects of genetic diversity that do not contribute to functional diversity or trophic interactions as measured in this study may reduce ecosystem functioning, for example, due to trade-offs between genetic diversity and species diversity. For example, it has been shown that in species-diverse grassland ecosystems, niche-complementarity between species can increase at the expense of reduced variation within species (*van Moorsel et al., 2018*; *van Moorsel et al., 2019*; *Zuppinger-Dingley et al., 2014*; *Zvereva et al., 2012*). Thus, our experiment simulating high genetic diversity within species in mixtures might have reduced the positive effects of high species diversity. This interpretation would be compatible with the observation that in the separate path-analyses diagrams direct negative effects of genetic diversity on productivity were only found in species mixtures, whereas in the species monocultures these effects were positive (see next section). Independent of this interpretation, our finding could also imply that partly different mechanisms underpin effects of species vs. genetic diversity on ecosystem functioning (*Barantal et al., 2019*; *Des Roches et al., 2018*).

## Effects of tree genetic diversity differ between tree species monocultures and mixtures

In contrast to our second hypothesis, we found that the effects of genetic diversity via functional diversity and soil fungal diversity were negative in species monocultures but not significant via functional diversity and positive via soil fungal diversity in the species mixture (*Figure 5*). We found that genetic diversity had positive effects on tree functional diversity and soil fungal diversity in species monocultures but negative effects in the species mixture, which supports the trade-offs between genetic and species diversity discussed in the previous section. However, the hypothesized positive effects of tree functional diversity on productivity turned negative in species monoculture. This result indicates that functional diversity may not have positive effects on the ecosystem functioning under low environmental heterogeneity, that is, species monocultures in our study (*Hillebrand and Matthiessen, 2009*). Moreover, other aspects of tree genetic diversity seem to play an important role not only for productivity in tree species mixtures (see previous section) but also for productivity in tree species monocultures. These may include unmeasured functional traits such as root traits (*Bardgett et al., 2014*) or unknown mechanisms underpinning effects of tree genetic diversity.

The two methods of calculating functional diversity either from seed-family means or from trait values of individual trees yielded different results regarding the indirect effects of genetic diversity on tree productivity via functional diversity. The method based on seed-family means has the advantage to be less circular, whereas the method based on trait values of individuals has the advantage of producing functional diversity values >0 also for genetic monocultures of single species (1.1 communities; see 'Materials and methods'). The weaker indirect effects of genetic diversity on tree productivity via functional diversity in the method using trait values of individuals suggest that the zero value of functional diversity in 1.1 communities in the method using seed-family means may lead to an overestimation of these indirect effects of genetic diversity in species monocultures. Nevertheless, the method using seed-family means is still useful for species monocultures with multiple seed families and for species mixtures.

## Conclusions

In this study, we tried to disentangle the effects of tree species and genetic diversity via functional diversity and trophic feedbacks on tree community productivity in a simple experimental system with four species and multiple seed families per species. Even though this was already challenging to set up, manage, and assess by measurements on trees and soil samples, larger studies will be required to generalize results. Nevertheless, our results suggest that both partitioning of resource-use and enemy niches (*Turnbull et al., 2016*) between and among genotypes within tree species played a role in affecting tree community productivity. Although both tree species and genetic diversity contributed to productivity, the underpinning mechanisms differed and were harder to explain for tree genetic diversity. We suggest that trade-offs between tree species and genetic diversity may cause the latter to switch strength and direction between species monocultures and mixtures. We were not able to definitively report causality between trophic feedbacks and tree productivity because we did not experimentally manipulate herbivore leaf damage and soil fungi. However, our results do support the hypothesis that trophic feedbacks affect plant community productivity. Given the importance of afforestation projects to mitigate carbon loss and provide ecological and economic benefits (*Brockerhoff et al., 2008*; *Lamb et al., 2005*), we strongly recommend that both tree species and genetic diversity should be considered in afforestation projects.

## Materials and methods
### Study site and experimental design

This study was carried out in the species × genetic diversity experiment of the BEF-China (https://www.bef-china.com; *Bruelheide et al., 2014*; *Hahn et al., 2017*). BEF-China is located close to Xingangshan, Dexing City, Jiangxi Province, China. The mean annual temperature is 16.7 °C, and the mean annual precipitation is 1821 mm. The species × genetic diversity experiment was established in 2010 and comprises 24 plots of 25.8 × 25.8 m equal to one Chinese unit of 'mu'. Each plot was planted with 400 individual trees from a pool of four species (*Alniphyllum fortunei*, *Cinnamomum camphora*, *Daphniphyllum oldhamii,* and *Idesia polycarpa*) with the mother trees of all tree individuals known (*Appendix 1—figure 1*). We defined the offspring from the same mother tree as a seed family and assumed that the genetic variation was larger among seed families than within a seed family (*Bongers et al., 2020*; *Hahn et al., 2017*). Since the offspring of a single mother tree could have been sired by different father trees, they represented anything between full- and half-sib families. Thus, in this study, we used the number of seed families per species as a measure of genetic diversity (*Bruelheide et al., 2014*). Across the 24 plots, we combined species diversity (one or four species) and genetic diversity (one or four seed families per species), which resulted in four tree diversity levels: one species with one seed family (1.1), one species with four seed families (1.4), four species with one seed family per species (4.1), and four species with four seed families per species (4.4) (*Appendix 1—figure 1*; *Bongers et al., 2020*).

For each of the four species, we collected seeds from eight mother trees to allow for two replications of four-family mixtures per species. Furthermore, to avoid the effects of unequal representation of particular seed families and correlations between seed family presence and diversity treatments, we made sure that every seed family occurred the same number of times at each diversity level (see *Appendix 1—table 1*, small deviations from the rule were required where not enough seeds from a

seed family could be obtained). Due to budget limitations and the number of replicates required per single seed family, the 1.1 and 1.4 diversity treatments were applied at subplot level (0.25 mu) and replicated 32 and 8 times, respectively. The 4.1 and 4.4 diversity treatments were applied at plot level (1 mu) and were replicated eight and six times, respectively (*Appendix 1—figure 1*; see also Figure 1 in *Bongers et al., 2020*). To allow for simpler analysis, we obtained most community measures at subplot level also for the 4.1 and 4.4 diversity treatments and thereafter used the subplots for all tests of diversity effects on these community measures, including plots as error (i.e., random-effects) term for testing the diversity effects in the corresponding mixed models. In total, because one 1-mu plot could not be established due to logistic constraints, the number of subplots used was 92 (32 subplots of 1.1, 8 subplots of 1.4, 28 subplots of 4.1, and 24 subplots of 4.4 diversity treatment). Note that in biodiversity experiments lower richness levels represent more different communities and thus require more plots. For the highest richness level, where there is typically only one species composition, this same community is typically replicated multiple times, as we did here for the 4.4 diversity treatment.

## Tree functional traits and functional diversity

Five leaf functional traits were measured in 2017 and 2018, including leaf area (LA), specific leaf area (SLA), chlorophyll content (CHL), leaf nitrogen content (LN), and leaf carbon content (LC). These traits can reflect the resource acquisition ability of plants and may show substantial variation not only among species but also within species (*Albert et al., 2010*; *Cornelissen et al., 2003*). We collected these traits on 547 individuals of all the seed families of the four species across all the species ×genetic diversity combinations (*Appendix 1—table 2*), with details described in *Bongers et al., 2020*.

Functional leaf-trait diversity was expressed as multivariate functional dispersion (FDis), which in our case corresponds to the mean distance of individual seed families to the centroid of all seed families in the community (*Laliberté and Legendre, 2010*). To reduce circularity, we used the seed-family means across all species × genetic diversity combinations to calculate FDis values per subplot that did not only depend on the functional trait measures obtained in that particular subplot. Using traits measured in a particular subplot to calculate FDis for that subplot bears the risk that the measured traits reflect a response to the local environment, yet we want to use FDis as a predictor variable for the performance of that subplot. In every mixture, trait values were weighted equally across seed families and species because these were planted in equal numbers in each subplot. The mean value of FDis per species × genetic diversity level was used to fill in missing values in a few subplots with families lacking trait data (*Appendix 1—table 2*). We also calculated another frequently used functional diversity index, Rao's Q (*Rao, 1982*). However, a strong positive correlation was detected between FDis and Rao's Q in simulated data (*Laliberté and Legendre, 2010*) and in our study (*Appendix 2—figure 2*). Moreover, in the case of equal weighting, FDis should perform better than Rao's Q (*Laliberté and Legendre, 2010*). Therefore, we only used FDis in the analyses presented in this study. The calculations of FDis and Rao's Q were done with the 'dbFD' function of the 'FD' package (versions 1.0–12.1) in R (*Laliberté et al., 2014*, https://www.r-project.org). We further calculated FDis using traits measured on individual trees across all tree diversity treatment combinations. This alternative FDis had the advantage that it could also be calculated for subplots planted with trees of a single seed family (which had FDis values of zero when calculated with seed-family means), reflecting within seed-family functional trait diversity. The disadvantage is that this measure likely includes more response variation because every individual tree responds to a number of unknown factors in its local environment. We also calculated CWMs for the five functional traits. To obtain a multivariate equivalent, we subjected the individual traits to a varimax rotation principal component analysis (PCA) to obtain two orthogonal axes as principal-component CWMs. The two principal components captured together 64% variation of trait variation (*Appendix 2—table 2*). PC1 indicated the functional traits directly connected with growth, and PC2 indicated the functional traits connected with photosynthesis (*Appendix 2—figure 3*). The varimax rotation PCA was done usinh 'psych' R package version 2.1.9 (*Makowski, 2018*).

## Trophic interactions

### Herbivory

Herbivory results from the interaction between plants and herbivores and can be recorded as leaf damage. For every individual tree, four or five damaged leaves were randomly collected and herbivory visually estimated (*Johnson et al., 2016*) (same 547 trees as for the traits, see above) in 2017. Thus,

in this study herbivory represents the percentage of damaged area per leaf attacked by herbivores. The herbivory caused by chewers, gall formers, leaf miners, and rollers were collectively counted. Because we only collected damaged leaves in this study, we might have overestimated the herbivory per individual tree. We therefore used data from other plots of the BEF-China experiment (*Schuldt et al., 2015*), which did not exclude nondamaged leaves to correct the potential bias. This former study assessed herbivore damage by visually inspecting 21 leaves (7 leaves per branch) on three random branches from different parts of the canopy (*Schuldt et al., 2015*). They used the mean percentage damage value as the overall leaf damage for each individual. We related leaf damage of corresponding tree individuals from this former study (total leaf damage) to leaf damage excluding nondamaged leaves (damage per damaged leaf) for all four species by linear regression (Pearson's correlation = 0.86–0.96, p < 0.001) (*Appendix 2—table 3*). With these regression models, we got the predicted values of herbivory for our study and used these predicted values in the final analyses. The mean value of herbivore damage per species × genetic diversity level was used to fill in missing values in a few subplots with tree individuals lacking herbivory data (*Appendix 1—table 2*).

## Soil fungal diversity

Soil fungal diversity was used as a proxy of unspecified trophic interactions. To be consistent with the species and genetic diversity treatment design, soil samples were taken on subplot level for the 1.1 and 1.4 diversity treatments, but, due to feasibility constraints, on plot level for the 4.1 and 4.4 diversity treatments in 2017. In each subplot or plot, five soil samples from the top 0–5 cm soil layer were collected from the four corners and the center of each subplot or plot. The five samples were then mixed together. Each soil sample was packed with dry ice and transferred to the laboratory for storage at −80°C until DNA extraction. The total genomic DNA of the subsample was extracted using Soil Genomic DNA Kit (Tiangen Biotech Co., Beijing, China), following the manufacturer's protocol. The DNA was extracted to perform PCR amplification. We amplified the nuclear rDNA internal transcribed spacer 2 (ITS2) region using primers ITS3F (GCATCGATGAAGAACGCAGC) and ITS4R (TCCTCCGC TTATTGATATGC). We processed the raw sequences with the QIIME 2 pipeline (https://docs.qiime2.org/) to cluster and assign operational taxonomic units (OTU). The fungal OTU tables were rarefied to 10,975 reads to account for the different sequencing depths. We then assigned the sequences to taxonomic groups using the UNITE database (*Nilsson et al., 2019*). Based on the taxonomic and abundance information of every subplot or plot, the Chao1 diversity index (*Chao, 1984*) was used to quantify soil fungal diversity, because most fungal species in our study were relatively rare and the Chao1 index can account well for rare species (*Chao, 1984*). The calculation of diversity of soil fungi was done using the 'vegan' package version 2.5–7 in R (*Oksanen et al., 2019*).

## Tree community productivity

We measured the basal area (BA) and the height (H) of all trees in the species × genetic diversity plots in 2018 (*Bongers et al., 2020*). Individual tree biomass (kg) was calculated using the biomass equation (H × BA × CV) of the BEF-China experiment (*Huang et al., 2018*) in which CV is a correction factor for stem shape and wood density. More details about the biomass equation can be found in *Huang et al., 2018*. We summed the biomass of individual trees to subplot level to calculate tree community productivity (Mg ha$^{-1}$).

## Statistical analysis

First, we evaluated the bivariate relationships between tree diversity, trophic interactions, and tree community productivity. To determine how species and genetic diversity and their interaction affected tree functional diversity and trophic interactions, linear mixed-effects models (LMMs) were fitted with two types of contrast coding. In the first, we used the ordinary two-way analysis of variance with interaction and in the second we replaced the genetic diversity main effect and the interaction with separate genetic diversity effects for species monocultures and the species mixture (*Appendix 2—table 4*). Note that as our design was orthogonal, fitting sequence did not matter in either of the codings. However, we focused our major analysis on the second type of coding to make it consistent with our hypotheses. Main effects of genetic diversity are presented in inset panels in *Figure 2*. Our second contrast coding ensured that we tested the effects of genetic diversity separately in species monocultures and species mixture, but within the same analysis. For all LMMs, we used 'plot' as a random

variable since subplots were nested in plots. This also ensured that fixed terms whose levels did not vary within plots among subplots (specifically the four-species diversity treatments) were correctly tested against the variation among plots rather than the residual variation among subplots. LMMs were fitted in R with the 'lmer' function of the lme4 package version 1.1.27.1 (*Bates et al., 2015*) using Kenward–Roger's method to calculate denominator degrees of freedom and F-statistics with the lmerTest-package version 3.1.3 (*Kuznetsova et al., 2017*). To meet the assumptions of linear mixed models, the proportion of leaf damage caused by herbivores was angular transformed (*Snedecor and Cochran, 1989*). For the display of regression lines in *Figure 3*, we used linear models relating tree functional diversity, herbivore leaf damage, and soil fungal diversity for the four diversity-treatment combinations to tree community productivity ('lm' function in R).

Second, we fitted SEMs and displayed the results in path-analysis diagrams (*Grace, 2006*) with the 'piecewiseSEM' package version 2.1.2 in R (*Lefcheck, 2016*) to assess causal hypotheses about how the effects of tree species and genetic diversity on community productivity could have been mediated via tree functional diversity and trophic interactions. The initial model was constructed by the most relevant pathways derived from theoretical assumptions (*Figure 4—figure supplement 2*). Additionally, we used separate linear regressions to assess the relationships between variables hypothesized to be related in cause–effect relationships in the SEMs. We assumed that both tree genetic diversity and species diversity could influence trophic interactions and community productivity directly or indirectly, that is, mediated via functional diversity (*Müller et al., 2018*; *Scherber et al., 2010*; *Schuldt et al., 2019*). Moreover, we hypothesized that tree functional diversity, herbivore leaf damage, and soil fungal diversity have direct feedbacks on community productivity (*Eisenhauer, 2012*; *Semchenko et al., 2018*). We sequentially dropped noninformative pathways if their removal reduced the AICc of the SEMs by more than 2 (*Grace, 2006*). To detect potential distorting effects of the relationship between genetic diversity and functional diversity calculated from seed-family means, we also calculated a SEM model without the path between genetic diversity and functional diversity.

Thirdly, separate multigroup SEMs were fitted for species monocultures and mixtures since significant interactions between species and genetic diversity in the ANOVAs indicated that genetic diversity had different effects between species monocultures and the species mixtures. The initial multigroup path diagram is shown in *Figure 5—figure supplement 2*. We simplified the multigroup initial model with the same procedure as described above by comparing AICc values. For the multigroup models, we also calculated an additional one in which the path between genetic diversity and functional diversity was excluded.

Finally, to detect the robustness of our results, we used the same paths as in the above final single and multipath models to analyze the data with FDis calculated with the trait measures of individual trees. All the analyses were carried out in R 4.0.5.

## Acknowledgements

We acknowledge the support of the BEF-China platform of the Zhejiang Qianjiangyuan Forest Biodiversity National Observation and Research Station. This study was financially supported by the National Natural Science Foundation of China (31870409), Strategic Priority Research Program of the Chinese Academy of Sciences (XDB31000000), and National Natural Science Foundation of China (32161123003). XL was supported by the Youth Innovation Promotion Association CAS (2019082). BS was supported by the University Research Priority Program Global Change and Biodiversity of the University of Zurich.

## Additional information

### Competing interests

Bernhard Schmid: Reviewing editor, *eLife*. The other authors declare that no competing interests exist.

## Funding

| Funder | Grant reference number | Author |
|---|---|---|
| National Natural Science Foundation of China | 31870409 | Ting Tang<br>Franca J Bongers<br>Xiaojuan Liu |
| Strategic Priority Research Program of the Chinese Academy of Sciences | XDB31000000 | Naili Zhang<br>Xiaojuan Liu |
| National Natural Science Foundation of China | 32161123003 | Naili Zhang<br>Yu Liang<br>Keping Ma<br>Xiaojuan Liu |
| Youth Innovation Promotion Association CAS | 2019082 | Xiaojuan Liu |

The funders had no role in study design, data collection and interpretation, or the decision to submit the work for publication.

## Author contributions

Ting Tang, Conceptualization, Data curation, Software, Formal analysis, Visualization, Methodology, Writing - original draft, Writing - review and editing; Naili Zhang, Data curation, Methodology; Franca J Bongers, Andrew L Hipp, Visualization, Writing - review and editing; Michael Staab, Walter Durka, Data curation, Methodology, Writing - review and editing; Andreas Schuldt, Data curation, Investigation, Methodology, Writing - review and editing; Felix Fornoff, Hong Lin, Baocai Han, Data curation; Jeannine Cavender-Bares, Methodology, Writing - review and editing; Shan Li, Resources, Data curation, Investigation; Yu Liang, Methodology; Alexandra-Maria Klein, Helge Bruelheide, Data curation, Writing - review and editing; Bernhard Schmid, Conceptualization, Formal analysis, Methodology, Writing - review and editing; Keping Ma, Conceptualization, Resources, Supervision, Funding acquisition, Validation, Project administration, Writing - review and editing; Xiaojuan Liu, Conceptualization, Resources, Data curation, Formal analysis, Supervision, Funding acquisition, Validation, Investigation, Methodology, Writing - original draft, Project administration, Writing - review and editing

## Author ORCIDs

Ting Tang ⓘ http://orcid.org/0000-0002-1145-0723
Felix Fornoff ⓘ http://orcid.org/0000-0003-0446-7153
Yu Liang ⓘ http://orcid.org/0000-0003-4259-6028
Xiaojuan Liu ⓘ http://orcid.org/0000-0002-9292-4432

## Decision letter and Author response

Decision letter https://doi.org/10.7554/eLife.78703.sa1
Author response https://doi.org/10.7554/eLife.78703.sa2

# Additional files

## Supplementary files
• MDAR checklist

## 1Data availability
All numerical data were used to generate the figures that have been deposited in Dryad.

The following dataset was generated:

| Author(s) | Year | Dataset title | Dataset URL | Database and Identifier |
|---|---|---|---|---|
| Liu X | 2022 | Gata from: Tree species and genetic diversity increase productivity via functional diversity and trophic feedbacks | http://dx.doi.org/10.5061/dryad.gf1vhhmqx | Dryad Digital Repository, 10.5061/dryadgf1vhhmqx |

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

## Appendix 1

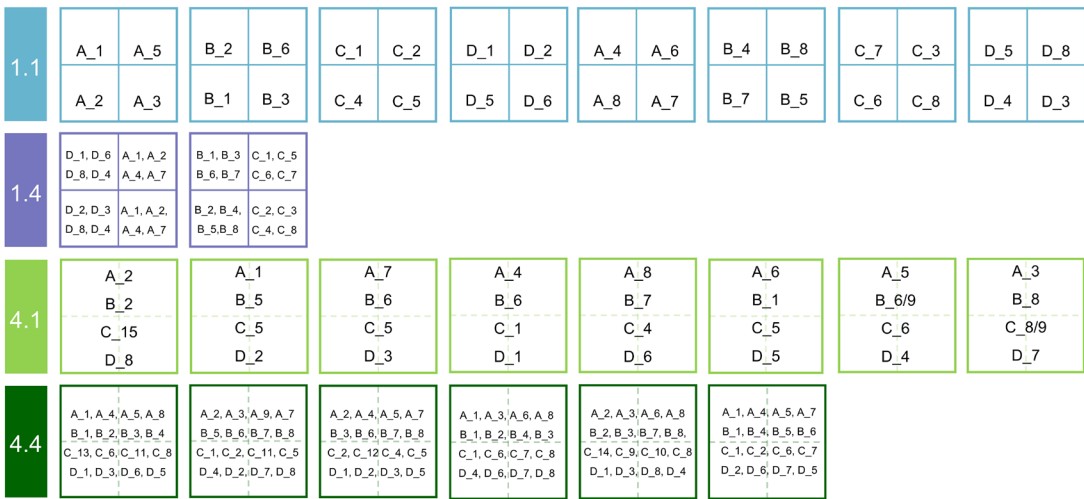

**Appendix 1—figure 1.** Diagram of the seed families planted in the species × genetic diversity experiment. 1.1: species diversity = 1, genetic diversity = 1; 1.4: species diversity = 1, genetic diversity = 4; 4.1: species diversity = 4, genetic diversity = 1; 4.4: species diversity = 4, genetic diversity = 4. The uppercase letters indicate four tree species (A: *Alniphyllum fortunei*; B: *Cinnamomum camphora*; C: *Daphniphyllum oldhamii*; D: *Idesia polycarpa*), the number after '_' indicates the seed family tag of a given species, two numbers indicate both of the seed families were used in this plot due to not enough designed seedlings.

**Appendix 1—table 1.** The designed and planted occurrence times of each seed family per species in the four diversity treatment combinations.

1.1: species diversity = 1, genetic diversity = 1; 1.4: species diversity = 1, genetic diversity = 4; 4.1: species diversity = 4, genetic diversity = 1; 4.4: species diversity = 4, genetic diversity = 4. 'SP' is the species name and 'SF' is the tag of seed family.The experiment was designed to use eight seed families per species, but additional or repeated seed families were used to complement the lack of enough individuals in some seed families. The numbers in brackets indicate the seed family tags that were used to complement. AlFo (A): *Alniphyllum fortune'*, CiCa (B): *Cinnamomum camphora*; DaOl (C): *Daphniphyllum oldhamii*; IdPo (D): *Idesia polycarpa*. 'x' represents the number of individuals per subplot, and the number of the 'x' represents the number of subplots.

| Tree diversity | | 1.1 (x = 100) | 1.4 (x = 25) | 4.1 (x = 100) | 4.4 (x = 25) |
|---|---|---|---|---|---|
| SP | SF | Tree individuals | Tree individuals | Tree individuals | Tree individuals |
| AlFo | 1 | x | x | x | x + x + x |
| (A) | 2 | x | x | x | x + x + x |
| | 3 | x | x(1) | x | x + x + x |
| | 4 | x | x | x | x + x + x |
| | 5 | x | x(2) | x | x + x + x |
| | 6 | x | x(4) | x | x + x + x(9) |
| | 7 | x | x | x | x + x + x |
| | 8 | x | x(7) | x | x + x + x |
| CiCa | 1 | x | x | x | x + x + x |
| (B) | 2 | x | x | x | x + x + x |
| | 3 | x | x | x(6) | x + x + x |

*Appendix 1—table 1 Continued on next page*

*Appendix 1—table 1 Continued*

| Tree diversity | | 1.1 (x = 100) | 1.4 (x = 25) | 4.1 (x = 100) | 4.4 (x = 25) |
|---|---|---|---|---|---|
| | 4 | x | x | x(6/9) | x + x + x |
| | 5 | x | x | x | x + x + x(3) |
| | 6 | x | x | x | x + x + x |
| | 7 | x | x | x | x + x + x |
| | 8 | x | x | x | x + x + x |
| DaOl | 1 | x | x | x | x + x + x |
| (C) | 2 | x | x | x(7) | x + x + x |
| | 3 | x | x | x(7) | x(11) + x(11) + x(9) |
| | 4 | x | x | x | x + x(10) + x(12) |
| | 5 | x | x | x | x + x + x(13) |
| | 6 | x | x | x | x + x + x |
| | 7 | x | x | x | x + x + x(14) |
| | 8 | x | x | x(8/9) | x + x + x |
| IdPo | 1 | x | x | x | x + x + x |
| (D) | 2 | x | x | x | x + x + x |
| | 3 | x | x | x | x + x + x |
| | 4 | x | x | x | x + x + x |
| | 5 | x | x(4) | x | x + x + x |
| | 6 | x | x | x | x + x + x |
| | 7 | x(5) | x(8) | x | x + x + x |
| | 8 | x | x | x | x + x + x |

**Appendix 1—table 2.** Data description of multi-trophic levels.

| Data type | Data description | Subplots | Year |
|---|---|---|---|
| Plant trait | LA, SLA, CHL, LN, LC | 77 | 2017 |
| Herbivore damage | Visually estimated | 77 | 2017 |
| Soil fungi | Mainly composed of saprophytes | 53 | 2017 |
| Community productivity | Sum of the biomass per subplot/area of subplot | 92 | 2018 |

LA, leaf area; SLA, specific leaf area; CHL, chlorophyll content; LN, leaf nitrogen content; LC, leaf carbon content.

# Appendix 2

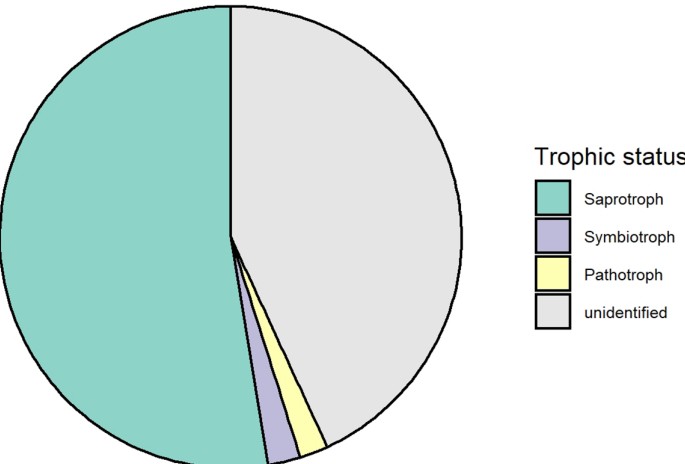

**Appendix 2—figure 1.** Trophic composition of soil fungi in this study. All fungi from this study were pooled together to calculate the relative abundance of each trophic group.

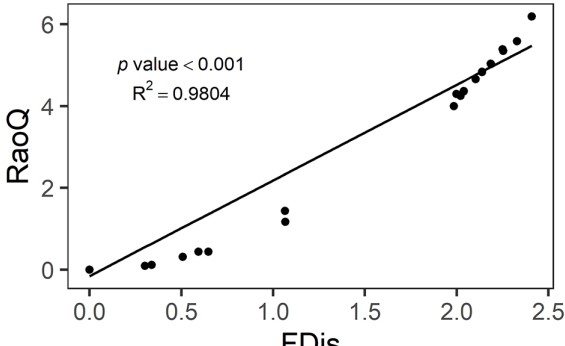

**Appendix 2—figure 2.** Relationship between functional dispersion (FDis) and Rao's Q (RaoQ).

**Appendix 2—table 1.** Summary of linear mixed-effects models (LMMs) of species diversity (SD), genetic diversity (GD), and their interactions on tree productivity, tree functional diversity, trophic interactions, and community-weighted mean (CWM) of functional traits.

Expressed values are Df representing degree of freedom and F-values with related significances, *** p < 0.001; * *p < 0.01; * p < 0.05, + p < 0.1. Note that the very small F-values for CWMs are due to the equal representation of seed families across all tree diversity treatments (*Appendix 1—table 1*).

| Factors | Df | Random | Tree productivity F-value | FD (Mean) F-value | FD (Individual) F-value | Herbivory F-value | Soil fungal diversity F-value | CWM (RC1) F-value | CWM (RC2) F-value |
|---|---|---|---|---|---|---|---|---|---|
| SD | 1 | Plot | 6.16* | 44.80*** | 20.60*** | 0.08 | 1.06 | 0.004 | 0.002 |
| GD | 1 | Plot | 0.51 | 3.66+ | 0.11 | 5.86* | 1.57 | 0.000 | 0.033 |
| SD × GD | 1 | Plot | 0.23 | 7.44* | 0.05 | 1.44 | 5.29* | 0.003 | 0.011 |
| SD | 1 | Plot | 6.16* | 44.80*** | 20.60*** | 0.08 | 1.06 | 0.004 | 0.002 |
| GD (Sp-mono) | 1 | Plot | 0.00 | 11.09** | 0.15 | 0.17 | 1.35 | 0.002 | 0.000 |
| GD (Sp-mix) | 1 | Plot | 0.74 | 0.00 | 0.02 | 7.13* | 5.51* | 0.001 | 0.044 |

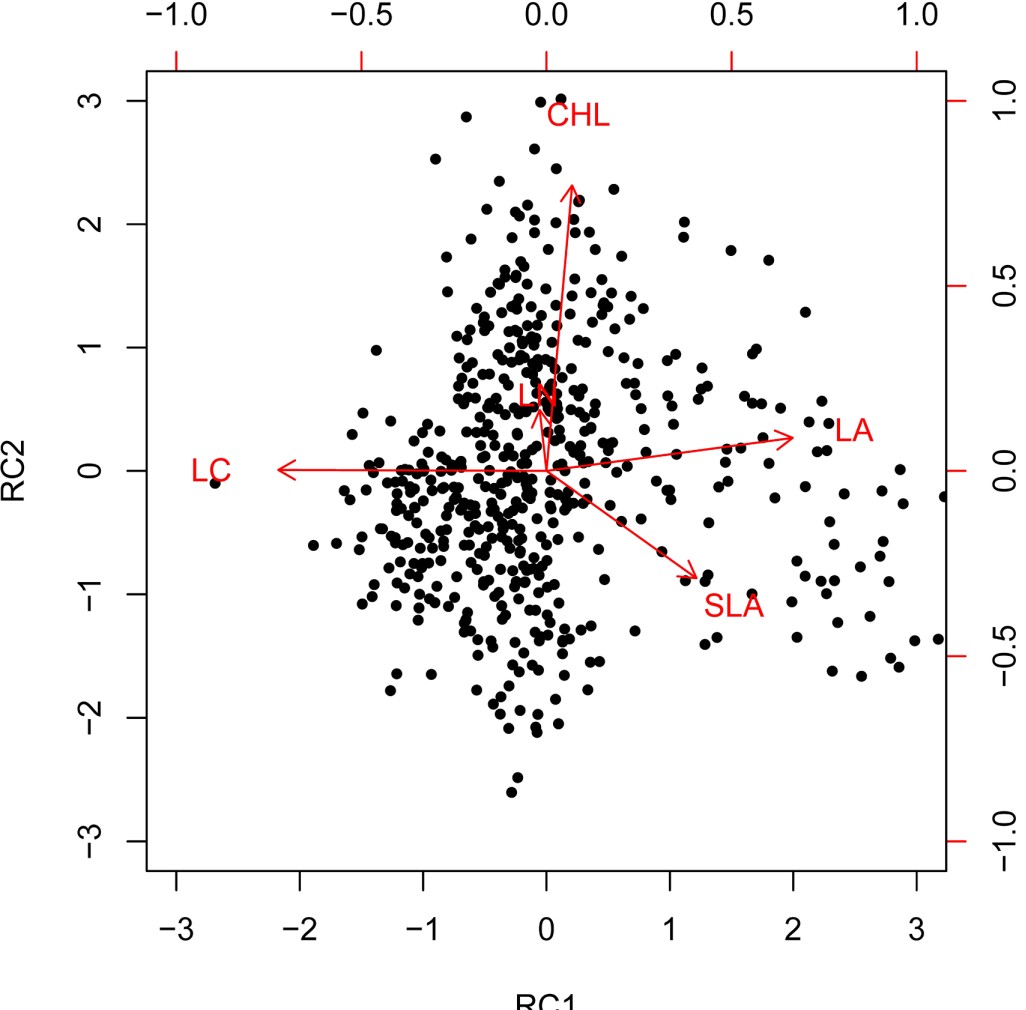

**Appendix 2—figure 3.** Varimax rotation principal component analysis (PCA) biplot for the five functional traits.

**Appendix 2—table 2.** Dimension reduction of community-weighted mean trait values (CWMs) by varimax rotation principal component analysis (PCA).

Loadings and eigenvalues of rotation principal components (RC) selected from a varimax rotation PCA on the CWM of leaf traits (most influential variables in bold).

|  | RC1 | RC2 |
|---|---|---|
| LA | **0.49** | 0.11 |
| SLA | **0.17** | −0.34 |
| CHL | 0.05 | **0.86** |
| LN | −0.20 | **0.16** |
| LC | −**0.55** | −0.01 |
|  |  |  |
| Explained | 41% | 23% |
| Cumulative explained | 41% | 64% |

LA, leaf area; SLA, specific leaf area; CHL, chlorophyll content; LN, leaf nitrogen content; LC, leaf carbon content.

**Appendix 2—table 3.** Results of linear models of leaf damage excluding undamaged leaves (this study) – leaf damage including undamaged leaves (from other plots of the BEF-China experiment) for the four species used in this study.

These models were used to correct the potential bias of herbivory estimates as a result of only collecting damaged leaves.

| Species | Slope | Intercept | $R^2$ | Pearson's correlation |
|---|---|---|---|---|
| *Alniphyllum fortunei* | 0.89970 | 2.83483 | 0.86 | 0.93 |
| *Cinnamomum camphora* | 0.94465 | 2.07484 | 0.73 | 0.86 |
| *Daphniphyllum oldhamii* | 0.88387 | 2.54032 | 0.92 | 0.96 |
| *Idesia polycarpa* | 0.92406 | 1.77523 | 0.86 | 0.93 |

**Appendix 2—table 4.** Contrast coding of genetic diversity in species monocultures and species mixtures separately.

Sp-mono presents species monocultures, and Sp-mix presents species mixtures.

| Species diversity | Genetic diversity | Genetic diversity in Sp-mono | Genetic diversity in Sp-mix |
|---|---|---|---|
| 1 | 1 | -1 | 0 |
| 4 | 1 | 0 | -1 |
| 1 | 4 | 1 | 0 |
| 4 | 4 | 0 | 1 |

**Appendix 2—table 5.** The interaction of significant results and the explained variance of the whole model of the multigroup structural equation models (SEM) shown in *Figure 5*.

\*\*\* p < 0.001; \*\* p < 0.01; \* p < 0.05.

| Response | Predictor | DF | Test.Stat | Explained variance % |
|---|---|---|---|---|
| Tree functional diversity | GD:SD | 1 | 0.5* | 11 |
| Herbivory | GD:SD | 1 | 0.0 | 11 |
| Herbivory | Tree functional diversity:SD | 1 | 0.0* | |
| Soil fungal diversity | GD:SD | 1 | 52982.7*** | 6 |
| Tree community productivity | GD:SD | 1 | 327.6*** | 24 |
| Tree community productivity | Herbivory:SD | 1 | 327.6 | |
| Tree community productivity | Soil fungal diversity:SD | 1 | 327.6 | |
| Tree community productivity | Tree functional diversity:SD | 1 | 327.6* | |

SD, species diversity; GD, genetic diversity.

## Appendix 3

To investigate the possible consequences of the zero functional diversity of genotype monocultures of single species (diversity treatment = 1.1), we further added SEM without the path between genetic diversity and functional diversity. We found that excluding the path between genetic diversity and functional diversity in the SEM models did not change the direction and significance of other paths. These results indicate possible artifacts brought in by the zero functional diversity in 1.1 communities do not affect the remaining effects that we found in the analyses.

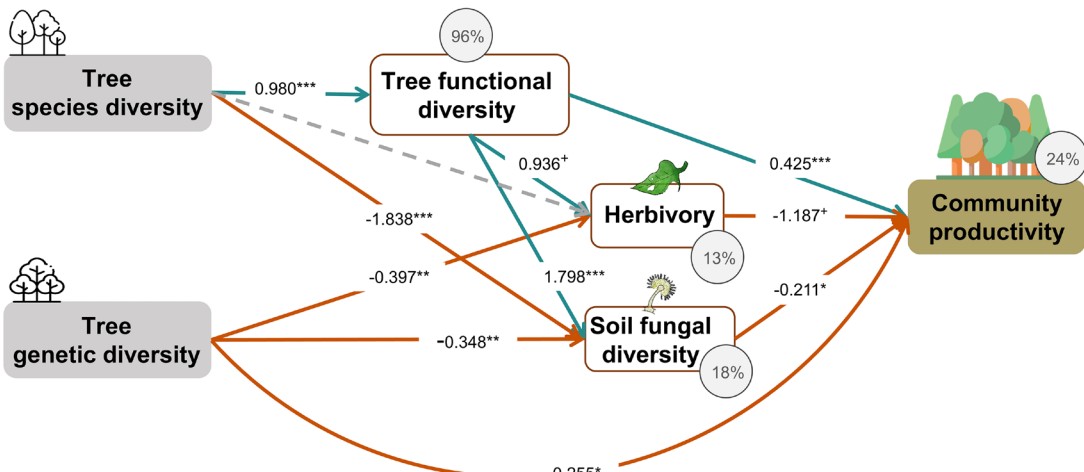

**Appendix 3—figure 1.** Effects of tree diversity on higher trophic levels and tree community productivity without the path between genetic diversity and functional diversity (global Fisher's C = 16.766, DF = 6, p = 0.01). Positive and negative paths are indicated in green and orange, respectively. The standardized path coefficients are indicated by the numbers, statistical significance is indicated by asterisks (*** p < 0.0001, ** p < 0.001, * p < 0.05, and + p <0.1), and the explained variance of dependent variables is indicated by the percentage values. The gray dashed line indicates a nonsignificant (p > 0.1) pathway in the final model. To allow comparison, the same structural equation model (SEM) as for *Figure 4* was used except excluding the path between genetic diversity and functional diversity.

**(a) Species monoculture**

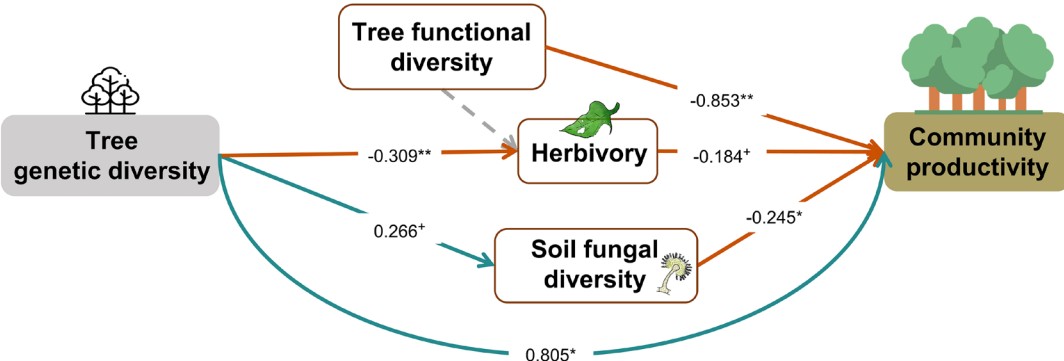

**(b) Species mixture**

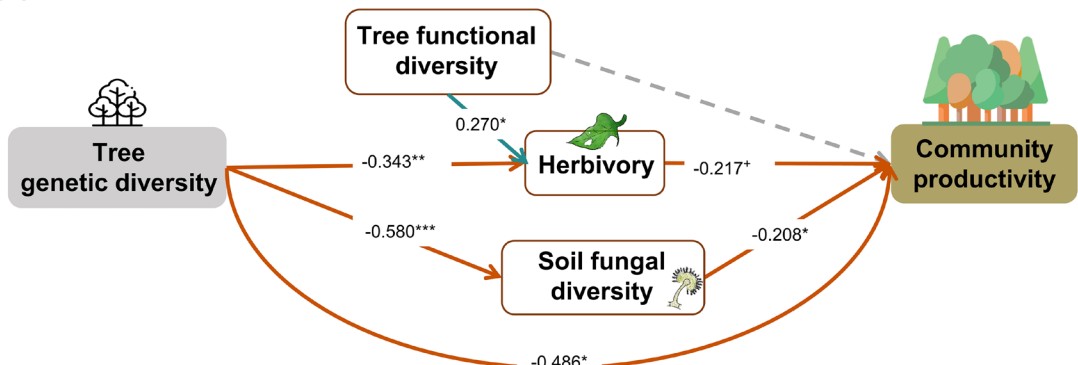

**Appendix 3—figure 2.** Effects of tree genetic diversity on higher trophic levels and tree community productivity in tree species monocultures (**a**) and mixtures of four tree species (**b**) without the paths between genetic diversity and functional diversity. The results were obtained by a multigroup structural equation model (SEM) (global Fisher's C = 3.485, DF = 4, p = 0.480). Positive and negative paths are indicated in green and orange, respectively. The standardized path coefficients are indicated by the numbers, and statistical significance is indicated by asterisks (*** p < 0.0001, ** p < 0.001, * p < 0.05, and + p <0.1). Gray dashed lines indicate nonsignificant (p > 0.1) pathways in the final model. Here, tree functional diversity was calculated from traits measured on individual trees. To allow comparison, the same SEM as for **Figure 5** was used. Multigroup SEM analyses first test the interaction (explanatory variable × groups) in the whole model using the full dataset and then estimate the local coefficient for each path by using different datasets (the full dataset or group sub-datasets [species richness = 1 or 4, respectively]) depending on the significance of explanatory variable × groups interactions. Thus, we could not get the percentage of the explained variance in the local multigroup SEM model.

