## [Editor Report]

This study uses a landmark experiment to provide compelling evidence that two mechanisms (increased trait space and biological interaction through herbivores and soil fungi) interact with intra- and interspecific genetic diversity to promote forest productivity. These results will be important to foresters and molecular ecologists looking to improve their practices to increase or maintain forest ecosystem functions.

---

## [Decision Letter]

**Decision letter after peer review:**

Thank you for submitting your article "Tree species and genetic diversity increase productivity via functional diversity and trophic feedbacks" for consideration by *eLife*. Your article has been reviewed by 2 peer reviewers, and the evaluation has been overseen by a Reviewing Editor and Detlef Weigel as the Senior Editor. The reviewers have opted to remain anonymous.

Essential revisions:

Both reviewers provide a large set of questions on the experimental design, data collection, and data analyses that need to be fully addressed in a new version and a response letter. Please notice that formal acceptance depends on the quality of the new version and therefore not guaranteed at the moment.

*Reviewer #1 (Recommendations for the authors):*

Figure 3 shows that for monocultures with one genotype there is no functional diversity, which does of course not fully reflect reality. One way around this would be to know the functional diversity in these monocultures, e.g. if several individuals were measured in each monoculture. However, based on the Methods description (L291), traits were not measured in each plot but randomly across plots. This is a shortcoming, given that plant traits are known to change dependent on their plant neighbourhood. Alternatively, I think these monocultures should be removed from this figure.

Beyond functional diversity, the functional traits per se might be relevant and would also allow to test potential explanations for productivity differences between monocultures of single genotypes. For example, Community Weighted Means of traits could be calculated and used as explanatory variables for ecosystem functioning. Multivariate CWMs are also possible, e.g. using ordination.

L175-176: This tends to be true based on Figure 4, but not based on Figure 5. These differences suggest to me that the differences in the impact of genetic diversity on different ecosystem functions are probably not very consistent and clear. I would therefore be cautious with the interpretation of these results.

Figure 5: Regarding the multi-group SEM not declared as such in the figure legend, but in the Methods (L381): I would assume that the relationships between herbivory and productivity as well as between soil fungal diversity and productivity would be the same independent of species monocultures and mixtures, meaning that those relationships could be fixed among the different groups (i.e. species monoculture vs species mixture), while all other relationships could vary between groups.

L186-187: This title is a bit bold and does not reflect all the results, namely the direct effect of genetic diversity on productivity.

*Reviewer #2 (Recommendations for the authors):*

Lines 80-84: The impact of enemies can also be reduced in a mixed community by the effects of plant trait variation, for example, higher trait variation among plants can affect herbivore performance and herbivory see: Wetzel W, et al. 2016. Nature 539 (7629), 425-427; Bustos-Segura C. 2017. Ecology Letters, 20(1), 87-97

Line 99: Cook-Patton et al. (2011) used herbaceous plants, not trees.

Lines 174-176: This is not clear, actually in Figures 2 and 5 we observe a clear effect of genetic diversity on functional diversity in monocultures but weak in mixed species.

Lines 188-190: Please rewrite this sentence, it is not clear what type of diversity it is referring to and some statements do not match the results. The effects of genetic and species diversity were different. For example, SD has a positive indirect effect (mediated through fungal diversity) on productivity and GD has positive indirect effects (mediated through herbivory and fungal diversity) and a negative direct effect.

Line 190: It is also not clear which positive direct effect this is referring to.

Lines 193: Also there could be trade-offs between defense induction and growth.

Lines 254-256: Afforestation or reforestation? In the abstract "reforestation" is used.

Line 264: What was the separation among plots? Is it the field as in Bongers, et al. 2020? It would be good to have the diagram here as well.

Line 268: How many families were used per plant species? If more than four were used, how were they selected for each plot?

Lines 276-279: The design is not clear. What was the replication unit, plot, or subplot? It cannot be a different one among treatments, since the scale of the replicate has to be the same to be comparable. In Bongers et al. 2020, it is shown that treatment 1.1 has the same species within the plot, and treatment 1.4 has different species within the plot. This would be problematic since treatment plots of 1.4 have already a mix of species not just genotypes. If 1.1 has the same species within the subplot but is different among subplots, then it is clear that the replication unit is the subplot, but this has to be explicitly explained.

Line 277: The replication for 1.4 does not seem good enough, how is this justified? How reliable are the results for this treatment with only two plots? In addition, in Bongers et al. 2020, these two plots are shown on the same side of the land, which increases the likelihood that this is a biased result.

Lines 315-318: How non-damaged leaves were included? It is not clear how this regression was performed and how each variable was collected.

Line 326: Were the soil samples collected at the subplot or plot level? If they were collected at different scales between treatments, it would not be something comparable.

Line 355: If explanatory terms were analysed sequentially, it means that genetic diversity was analysed after the effects of species diversity were taken into account. How would an ANOVA type II differ from a sequential analysis? This could be justified in the methods, but also emphasize in the results/discussion, to allow for correct interpretation.

Figure 1c. Better name mother trees 1, 2, 3, and 4 to match the seed families naming.

Figure 2. Why the direct link between species diversity and productivity is not shown? Please, clarify this in the figure or methods. In addition, the paths could be changed to more contrasting colors to facilitate the reading for color-blind people.

[Editors’ note: further revisions were suggested prior to acceptance, as described below.]

Thank you for resubmitting your work entitled "Tree species and genetic diversity increase productivity via functional diversity and trophic feedbacks" for further consideration by *eLife*. Your revised article has been evaluated by Detlef Weigel (Senior Editor) and a Reviewing Editor.

The manuscript has been improved but there are some remaining issues that need to be addressed, as outlined below:

Both reviewers find that using species family means for trait values is problematic (for example, the assigning of zero values in Rao's Q calculations seems puzzling). While you have made a case for continuing to use family means in your R1 response, we find that a new opportunity must be given to sustain using these data, and explain possible artifacts when using zero values in, for example, your SEM paths. Possible alternatives (such as a SEM without the tests/paths between genotypic diversity and FDis due to colinearity issues when using family means) should be discussed (in an Appendix?), and the Discussion should also be extended, to make it clear this is a shortcoming of your analysis.

*Reviewer #1 (Recommendations for the authors):*

I have reviewed this revised version of the manuscript after already reviewing the original version (reviewer 1). In my opinion, the manuscript significantly improved and is much clear now. In particular, the detailed comments of reviewer 2 must have helped to clarify the methods, which in turn helped to better understand the results and discussion. I am overall happy with the analyses and also appreciate very much the inclusion of the results with traits measured at the individual level. I understand the reason of the authors with respect to circularity, but also maintain my concerns with using species family means, that do not only eliminate the potential responses of the genotypes to the treatments (which was the goal of the authors with using species family means) but also the potential effects of the genotypes. It is basically impossible to eliminate the response of a trait without also eliminating its effect. And in this study, I think this is particularly tricky because the results turn out to be significantly different when using species family means versus individual values, with important implications for the objectives of this study. And here I am in serious doubt whether the way the authors decided to go forward is the most appropriate one. I explain this based on the key result affected:

L263ff: I find the results of negative genotype diversity effects on productivity in species monocultures puzzling. And it, in fact, only appears in the SEM and only when using the species family means of trait values but not the measured trait values in each subplot. I can't really make a conclusion out of this, but it seems weird to me, and I really wonder to what extent this is due to the assumed 0 functional diversity of genotype monocultures (1.1 communities). I can't help but think this is an artifact. Actually, the SEM with the functional traits measured on individual trees matches much better the results observed in the binary analyses, where genotype diversity doesn't have an impact on functional diversity of species monocultures nor community productivity of species monocultures (Figure 2 and its supplement 1). It seems rather that increased functional diversity in species monocultures goes along with reduced productivity (Figure 3), which is, however independent of tree genetic diversity (as there is no clear relationship between tree genetic diversity and functional diversity of species monocultures as stated above – when individual trait values are used). So, it seems to me that the negative effect of tree genetic diversity on productivity in species monocultures, as claimed by the authors, is not a genetic diversity effect but a functional diversity effect independent of genetic diversity. That would at least be my interpretation of the results.

*Reviewer #2 (Recommendations for the authors):*

I thank the authors for thoroughly answering all the reviewers' questions and comments. I think the present version clarifies all the points raised and shows more clarity in the design and discussion.

I am glad to read this revised version and congratulate the authors for such an impressive and interesting study that represent a step forward in our understanding of the effects of plant diversity.

---

## [Author Response]

Reviewer #1 (Recommendations for the authors):Figure 3 shows that for monocultures with one genotype there is no functional diversity, which does of course not fully reflect reality. One way around this would be to know the functional diversity in these monocultures, e.g. if several individuals were measured in each monoculture. However, based on the Methods description (L291), traits were not measured in each plot but randomly across plots. This is a shortcoming, given that plant traits are known to change dependent on their plant neighbourhood. Alternatively, I think these monocultures should be removed from this figure.Beyond functional diversity, the functional traits per se might be relevant and would also allow to test potential explanations for productivity differences between monocultures of single genotypes. For example, Community Weighted Means of traits could be calculated and used as explanatory variables for ecosystem functioning. Multivariate CWMs are also possible, e.g. using ordination.

Thank you for the valuable comments. We fully understand plant traits will change depending on their neighborhood. However, in our experiment, we hypothesized that functional diversity would be an explanatory variable that was used to explain how species and genetic diversity will affect multi-trophic interactions and community productivity. In this case, if we calculate functional diversity only based on the individuals planted in a particular subplot, trait plasticity in response to the neighborhood will be involved (see Sapijankas et al., 2014; Niklaus et al., 2017), which makes the analyses partly circular. In theory, we should use the mean trait value from common gardens to calculate functional diversity, but here we use the mean value across all the plots in our experiment to represent the realistic forest ecosystem scenery.

Lines 339–344: “To reduce circularity, we used the seed-family means across all species × genetic diversity combinations to calculate FDis values per subplot that did not only depend on the functional trait measures obtained in that particular subplot. Using traits measured in a particular subplot to calculate FDis for that subplot bears the risk that the measured traits reflect a response to the local environment, yet we want to use FDis as a predictor variable for the performance of that subplot.”

The zero functional diversity in species and genetic monoculture comes from using the mean trait value of each seed family of each species to calculate functional diversity. This method has been typically used in other studies (e.g. Craven et al., 2018; Williams et al., 2017), and we believe that it provides a proper estimation of functional diversity. However, we agree that the “real” functional diversity would not be zero even in the species and genetic monoculture, so now we have added functional diversity calculated by traits measured on individual trees across all subplots and put the results into the supplementary figures (Figure 2—figure supplement 1, Figure 3—figure supplement 1, Figure 4—figure supplement 1, Figure 5—figure supplement 1). With the “individual” functional diversity method we could calculate FDis also for subplots planted with trees of a single seed family (which had FDis values of zero when calculated with seed-family means). Overall, the results from the “individual” method still support our hypotheses that tree species diversity and genetic diversity affect community productivity via tree functional diversity and multi-trophic feedbacks (Figure 4—figure supplement 1, Figure 5—figure supplement 1). Although some effects of functional diversity disappeared or were weaker with FDis based on individuals than on seed-family means, this is understandable because FDis based on individuals includes more environmental response variation among individuals which can blur the effects of tree genetic diversity and species diversity. We added this FDis calculation to the Methods, reported the different results from the two FDis methods, and justified the weaker effects of functional diversity by this analysis in the Discussion.

Lines 354–359: “We further calculated FDis using traits measured on individual trees across all tree diversity treatment combinations. This alternative FDis had the advantage that it could also be calculated for subplots planted with trees of a single seed family (which had FDis values of zero when calculated with seed-family means), reflecting within seed-family functional trait diversity. The disadvantage is that this measure likely includes more response variation because every individual tree responds to a number of unknown factors in its local environment.”

Lines 130–133: “However, when we calculated functional diversity based on measurements taken on individual trees rather than based on seed-family means, only species diversity but not genetic diversity had effects on tree functional diversity (Figure 2—figure supplement 1), indicating additional within seed-family variation masking some of the between seed-family variation.”

Lines 140–143: “Tree functional diversity calculated using either seed-family means or individual-tree values had positive overall effects on community productivity but this effect was mainly due to an increase in functional diversity from species monocultures to mixtures (Figure 3a, Figure 3—figure supplement 1).”

Lines 164–170: “The analysis which functional diversity calculated from measurements on individual trees also showed that tree species diversity and genetic diversity affect community productivity via tree functional diversity and trophic feedbacks, although the effects of functional diversity were less pronounced (Figure 4—figure supplement 1), possibly because functional diversity calculated from individual trees included more response functional diversity (Sapijanskas et al. 2014) than did functional diversity calculated form seed-family means.”

Lines 176–178: “Similar results were found in the analyses using functional traits calculated from measurements on individual trees, except that the effects of functional diversity on tree productivity became positive (Figure 5—figure supplement 1).”

Lines 226–233: “To account for possible effects of functional diversity within seed families, we also calculated functional diversity based on measurements of individual trees (see Methods). Overall, the results from this novel method still support our hypotheses that tree species diversity and genetic diversity affect community productivity via tree functional diversity and multi-trophic feedbacks (Figure 4—figure supplement 1), although compared with the typically-used “mean” method, the novel method includes more variation among individuals which partly reflects responses of traits to the particular local environment (Sapijanskas et al. 2014); and this may have blurred the mean effects of tree genetic diversity and species diversity (Figure 4, Figure 4—figure supplement 1).”

We agree that community weighted means (CWM) of traits could also be an explanatory variable for ecosystem functioning. However, in our experiment, due to the equal representation of seed families across tree diversity treatments (Appendix 1—table 1), we assume that there must be no effects of tree species and genetic diversity on community weighted means (CWMs) of tree functional traits. Now we have calculated CWM and statistically proved that there are no significant effects of tree diversity on CWM (Appendix 2—table 1; actually, the corresponding F-values are so small that they can only be explained with the assumed constancy due to equal representation of seed families across diversity treatments).

Lines 359–366: “We also calculated community weighted means (CWMs) for the five functional traits. To obtain a multivariate equivalent we subjected the individual traits to a varimax rotation principal component analysis (PCA) to obtain two orthogonal axes as principal-component CWMs. The two principal components captured together 64% variation of trait variation (Appendix 2—table 2). PC1 indicated the functional traits directly connected with growth and PC2 indicated the functional traits connected with photosynthesis (Appendix 2—figure 3). The varimax rotation principal component analysis was done by ‘psych’ R package version 2.1.9 (Makowski, 2018).”

Lines 136–139: “Due to the equal representation of seed families across tree diversity treatments (Appendix 1—table 1), we did not find any significant effects of tree species and genetic diversity effects on community weighted means (CWMs) of tree functional traits (Appendix 2—table 1).”

L175-176: This tends to be true based on Figure 4, but not based on Figure 5. These differences suggest to me that the differences in the impact of genetic diversity on different ecosystem functions are probably not very consistent and clear. I would therefore be cautious with the interpretation of these results.

In this sentence, we only wanted to state the relative strength of effects of species diversity and genetic diversity which could be derived from Figure 4, which we now pointed out more clearly in the text. Meanwhile, we have further discussed the different effects of genetic diversity via functional diversity on productivity between species monocultures and mixtures in the following discussion.

Lines 190–194: “Nevertheless, compared with effects of species diversity, effects of genetic diversity on tree community productivity through functional diversity were weaker, whereas effects of genetic diversity on trophic interactions were strong (see Figure 4), indicating that mechanisms underpinning effects of genetic diversity may in part differ from those underpinning effects of species diversity, as we will discuss below.”

Lines 256–264: “However, the hypothesized positive effects of tree functional diversity on productivity turned negative in species monoculture. This result indicates that functional diversity may not have positive effects on the ecosystem functioning under low environmental heterogeneity, i.e. species monocultures in our study (Hillebrand and Matthiessen 2009). Therefore, our findings show that the different effects of genetic diversity on tree productivity between species monocultures and mixtures not only depend on the different effects of genetic diversity on functional diversity and trophic interaction, but also on the varied tree productivity consequences from functional diversity and trophic interaction on tree productivity between species monocultures and mixtures.”

Figure 5: Regarding the multi-group SEM not declared as such in the figure legend, but in the Methods (L381): I would assume that the relationships between herbivory and productivity as well as between soil fungal diversity and productivity would be the same independent of species monocultures and mixtures, meaning that those relationships could be fixed among the different groups (i.e. species monoculture vs species mixture), while all other relationships could vary between groups.

In our study, we hypothesized that all the paths are possible to be different between species monocultures and species mixtures. So, we did not constrain any path beforehand, the SEM results showed the differences between species monocultures and species mixtures according to the data itself. Now we have declared this in the figure legend and added a table in the Appendix (Appendix 2—table 5) to show the interaction results tested from the multi-group SEM model.

Lines 822–825: “All the paths were allowed to be different between species monocultures and mixtures (none of the paths was constrained manually beforehand); the interaction statistics of the multi-group model are shown in Appendix 2—table 5.”

L186-187: This title is a bit bold and does not reflect all the results, namely the direct effect of genetic diversity on productivity.

We did not include the “direct effects” of genetic diversity on productivity in this title because we could not test if the “direct” effects in the SEM are real direct effects or just because of some other mediating factors which we did not include in this study. Thus, we interpreted the direct effects as the variation of plant community productivity we failed to explain by functional diversity and trophic feedbacks we included in this study instead of a clear result. However, we agree that the title of the section was a bit bold and thus inserted the word “mainly” in front of “via functional diversity and trophic feedbacks” (line 203).

Lines 234–242: “Even after accounting for tree functional diversity and trophic feedbacks, we still detected a direct negative effect of tree genetic diversity on tree productivity, while the direct effect of tree species diversity was fully explained by functional diversity and trophic feedbacks. This suggests that aspects of genetic diversity that do not contribute to functional diversity or trophic interactions as measured in this study may reduce ecosystem functioning, e.g. due to trade-offs between genetic diversity and species diversity. For example, it has been shown that in species-diverse grassland ecosystems niche-complementarity between species can increase at the expense of reduced variation within species (van Moorsel et al., 2018; van Moorsel et al., 2019; Zuppinger-Dingley et al., 2014; Zvereva et al., 2012).”

Reviewer #2 (Recommendations for the authors):Lines 80-84: The impact of enemies can also be reduced in a mixed community by the effects of plant trait variation, for example, higher trait variation among plants can affect herbivore performance and herbivory see: Wetzel W, et al. 2016. Nature 539 (7629), 425-427; Bustos-Segura C. 2017. Ecology Letters, 20(1), 87-97

Thank you very much for recommending these two important papers, now we have added this mechanism and the references to the main text.

Lines 77–81: “Trophic feedbacks can enhance the performance of species or genotype mixtures either by reducing herbivore damage though enhancing the diversity of nutrient traits (Wetzel et al., 2016) and chemical traits (Bustos-Segura et al., 2017) or enhancing diversity of beneficial mutualists (e.g. mycorrhizal fungi (Semchenko et al., 2018)) (Figure 1b).”

Line 99: Cook-Patton et al. (2011) used herbaceous plants, not trees.

Thank you for pointing out this reference mistake, now we have replaced the reference with an experimental study (Abdala-Roberts et al., 2015) that used trees to compare the importance of species diversity and genetic diversity.

Lines 94–98: “Although there are few forest experimental studies in which species and genetic diversity are simultaneously manipulated, most of them only compared their relative importance on ecosystem functions (Abdala-Roberts et al., 2015; Koricheva and Hayes, 2018), and we barely know their interactive effects via functional diversity and trophic feedbacks on plant productivity.”

Lines 174-176: This is not clear, actually in Figures 2 and 5 we observe a clear effect of genetic diversity on functional diversity in monocultures but weak in mixed species.

Thank you for pointing out the unclear description. Now we have rewritten this sentence and made clear that here compared with the effects of species diversity, the effects of genetic diversity on tree community productivity through functional diversity were weaker. Additionally, we have discussed the different effects of genetic diversity via functional diversity between species monocultures and mixtures in the following discussion.

Lines 190–194: “Nevertheless, compared with effects of species diversity, effects of genetic diversity on tree community productivity through functional diversity were weaker, whereas effects of genetic diversity on trophic interactions were strong (see Figure 4), indicating that mechanisms underpinning effects of genetic diversity may in part differ from those underpinning effects of species diversity, as we will discuss below.”

Lines 256–264: “However, the hypothesized positive effects of tree functional diversity on productivity turned negative in species monoculture. This result indicates that functional diversity may not have positive effects on the ecosystem functioning under low environmental heterogeneity, i.e. species monocultures in our study (Hillebrand and Matthiessen 2009). Therefore, our findings show that the different effects of genetic diversity on tree productivity between species monocultures and mixtures not only depend on the different effects of genetic diversity on functional diversity and trophic interaction, but also on the varied tree productivity consequences from functional diversity and trophic interaction on tree productivity between species monocultures and mixtures.”

Lines 188-190: Please rewrite this sentence, it is not clear what type of diversity it is referring to and some statements do not match the results. The effects of genetic and species diversity were different. For example, SD has a positive indirect effect (mediated through fungal diversity) on productivity and GD has positive indirect effects (mediated through herbivory and fungal diversity) and a negative direct effect.

Thank you for pointing out the unclear place. Here we only discuss the effects of species diversity and genetic diversity on productivity and trophic interactions via functional diversity; we have now rewritten this sentence.

Lines 205–213: “Although only species diversity but not genetic diversity was found to affect tree productivity in binary analyses, both kinds of diversity positively affected tree community productivity and trophic interactions via functional diversity according to our structural equation models (SEMs) depicted in the corresponding path-analysis diagrams (see Figure 4). Tree functional diversity appeared to enhance complementary resource acquisition at community level (Kahmen et al., 2006; Marquard et al., 2009; Williams et al., 2017), which consequently enhanced tree community productivity. Meanwhile, tree functional diversity also provided more niche opportunities to benefit generalist herbivores and soil fungi which reduced tree community productivity, as has been found for these tree species in a parallel field study nearby (Brezzi et al., 2017).”

Line 190: It is also not clear which positive direct effect this is referring to.

We now have specified the “positive direct effect” is the effect of tree functional diversity on tree community productivity.

Lines 208–211: “Tree functional diversity appeared to enhance complementary resource acquisition at community level (Kahmen et al., 2006; Marquard et al., 2009; Williams et al., 2017), which consequently enhanced tree community productivity.”

Lines 193: Also there could be trade-offs between defense induction and growth.

Thank you for this useful comment. Now we have added this statement in this part of the Discussion.

Lines 213–216: “It is expected that herbivory has negative effects on plant productivity via the reduction of leaf area (Zvereva et al., 2012) and photosynthesis of remaining leaves (Nabity et al., 2009), and via trade-offs between growth and herbivore defense (Züst and Agrawal, 2017).”

Lines 254-256: Afforestation or reforestation? In the abstract "reforestation" is used.

Thank you very much for pointing out the inconsistent words. Now we have changed all of them into “afforestation” since we do not refer to replanting of a forest as it has been before in this study.

Line 264: What was the separation among plots? Is it the field as in Bongers, et al. 2020? It would be good to have the diagram here as well.

Thank you very much for this suggestion. Now we have added a plot design diagram in the Appendix (Appendix 1—figure 1) which shows the species and seed families used in each plot. Additionally, we have added a table to better describe how many individuals of how many seed families we planted in our experiment (Appendix 1—table 1). The field is as in Bongers et al. (2020), Figure 1, which we now cite specifically in the Methods (line 314) and include here for convenience:

The side length or each plot is about 26 m (square root of 1 mu=1/15 ha), thus form the figure it can be seen that distances between plots of the same treatment were mostly more than 50 m, but distance was not controlled because treatments were allocated randomly.

Line 268: How many families were used per plant species? If more than four were used, how were they selected for each plot?

In total, we used eight seed families for every species and we randomly selected the seed-family combination for different tree-diversity combinations, and made sure that every seed family occurred the same number of times at each diversity level. Now we have added this information to the Methods and a table (Appendix 1—table 1) to explain that.

Lines 309–314: “For each of the four species, we collected seeds from eight mother trees to allow for two replications of four-family mixtures per species. Furthermore, to avoid the effects of unequal representation of particular seed families and correlations between seed family presence and diversity treatments we made sure that every seed family occurred the same number of times at each diversity level (see Appendix 1—table 1, small deviations from the rule were required where not enough seeds from a seed family could be obtained).”

Lines 276-279: The design is not clear. What was the replication unit, plot, or subplot? It cannot be a different one among treatments, since the scale of the replicate has to be the same to be comparable. In Bongers et al. 2020, it is shown that treatment 1.1 has the same species within the plot, and treatment 1.4 has different species within the plot. This would be problematic since treatment plots of 1.4 have already a mix of species not just genotypes. If 1.1 has the same species within the subplot but is different among subplots, then it is clear that the replication unit is the subplot, but this has to be explicitly explained.

As the answer to the above two questions, now we have added the plot design in the Appendix to better present the experimental design. We have different units for species monocultures and species mixtures due to budget limitations and the number of replicates required per single seed family. For treatment 1.4, the replication unit is subplot (0.25 mu).

We agree that ideally we would have wanted to grow all experimental communities in plots of 400 trees, but due to the difficulty to obtain enough seeds from a single family we used subplots. To ensure that diversity treatments were still the same at plot level, we grouped different family identities of the same species and different species identities for treatments 1.1 and 1.4, respectively. The first was to make sure we still had species monocultures per plot, which was not necessary for the second. Admittedly, it would have been better to spread diversity treatment 1.4 over more plots, but that would have required to plant subplots among the plot grid or to combine them with other treatments in neighboring subplots, and still would then not have allowed us to have diversity treatment 1.4 at plot level, a requirement which we have achieved with the current design. For testing effects of diversity treatments we of course used plot as error (random-effects term in the mixed models), so these subplots of diversity treatments 1.1 and 1.4 were not used as pseudoreplicates.

We have added more detail to the Methods, but not as much as provided in this reply as we believe it would make the description unneccessarily complicated. However, we could of course add more if requested.

Lines 314–322: “Due to budget limitations and the number of replicates required per single seed family, the 1.1 and 1.4 diversity treatments were applied at subplot level (0.25 mu) and replicated 32 and 8 times, respectively. The 4.1 and 4.4 diversity treatments were applied at plot level (1 mu) and were replicated 8 and 6 times, respectively (Appendix 1—figure 1; see also Figure 1 in Bongers et al., 2020). To allow for simpler analysis, we obtained most community measures at subplot level also for the 4.1 and 4.4 diversity treatments and thereafter used the subplots for all tests of diversity effects on these community measures, including plots as error (i.e. random-effects) term for testing the diversity effects in the corresponding mixed models.”

Lines 429–433 “For all linear mixed-effects models, we used ‘plot’ as a random variable since subplots were nested in plots. This also ensured that fixed terms whose levels did not vary within plots among subplots (specifically the four diversity treatments) were correctly tested against the variation among plots rather than the residual variation among subplots.”

Line 277: The replication for 1.4 does not seem good enough, how is this justified? How reliable are the results for this treatment with only two plots? In addition, in Bongers et al. 2020, these two plots are shown on the same side of the land, which increases the likelihood that this is a biased result.

As we mentioned in the answer to the above question, due to budget limitations and the number of replicates required per single seed family, we used subplot as a replication unit for the 1.4 treatment (Appendix 1—figure 1), so here we have 8 replications for the 1.4 treatment. As we mentioned before, the plot distribution is designed to be completely random in the experiment site which is independent of the diversity treatments. Although the two 1.4 plots by chance were located only a bit more than 100 m apart, there was nothing we could do about this, and we believe that the distance was large enough to avoid biased results. As explained above, the reason for grouping the 1.4 subplots of different species in two single plots was to have this treatment also represented at plot scale.

Lines 314–316: “Due to budget limitations and the number of replicates required per single seed family, the 1.1 and 1.4 diversity treatments were applied at subplot level (0.25 mu) and replicated 32 and 8 times, respectively.”

Lines 315-318: How non-damaged leaves were included? It is not clear how this regression was performed and how each variable was collected.

We used a data set in which all the leaves were sampled (both damaged leaves and non-damaged leaves) from the same experimental site. Then we established linear models from this dataset to relate the herbivore damage sampled from both damaged leaves and non-damaged leaves with the herbivore damage only sampled from damaged leaves. We found strong correlations of the herbivore damage results from the two sampling methods, so we used the linear models to correct the potential bias of our data which only sampled the damaged leaves. Now we have rewritten the description in the Methods section.

Lines 375–385: “Because we only collected damaged leaves in this study, we might have overestimated the herbivory per individual tree. We therefore used data from other plots of the BEF-China experiment (Schuldt et al., 2015) which did not exclude non-damaged leaves to correct the potential bias. This former study assessed herbivore damage by visually inspecting 21 leaves (7 leaves per branch) on three random branches from different parts of the canopy (Schuldt et al., 2015). They used the mean percentage damage value as the overall leaf damage for each individual. We related leaf damage of corresponding tree individuals from this former study (total leaf damage) to leaf damage excluding non-damaged leaves (damage per damaged leaf) for all four species by linear regression (Pearson's correlation = 0.86–0.96, P < 0.001) (Appendix 2—table 3). With these regression models, we got the predicted values of herbivory for our study and used these predicted values in the final analyses.”

Line 326: Were the soil samples collected at the subplot or plot level? If they were collected at different scales between treatments, it would not be something comparable.

As mentioned in the answer to the plot design, we collected soil samples based on the species and genetic diversity treatment-replication unit and we also used “plot” as a random factor when we tested the effects of species diversity and genetic diversity on soil fungal diversity (see above answers) to solve this problem from sampling at different scales between treatments. Now we have clarified this in the Methods section.

Lines 390–393: “Soil fungal diversity was used as a proxy of unspecified trophic interactions. To be consistent with the species and genetic diversity treatment design, soil samples were taken on subplot level for the 1.1 and 1.4 diversity treatments, but on plot level for the 4.1 and 4.4 diversity treatments in 2017.”

Lines 429–433: “For all linear mixed-effects models, we used ‘plot’ as a random variable since subplots were nested in plots. This also ensured that fixed terms whose levels did not vary within plots among subplots (specifically the four diversity treatments) were correctly tested against the variation among plots rather than the residual variation among subplots.”

Line 355: If explanatory terms were analysed sequentially, it means that genetic diversity was analysed after the effects of species diversity were taken into account. How would an ANOVA type II differ from a sequential analysis? This could be justified in the methods, but also emphasize in the results/discussion, to allow for correct interpretation.

Because this was a designed experiment with orthogonally crossed species and genetic diversity treatments, ANOVA type I and II yield the same results (very slight differences due to missing values). However, we used a different approach, as indicated above, to the typical two-way ANOVA with interaction. Instead of fitting genetic diversity and interaction with fitted genetic diversity separately for species monocultures and the species mixture (note that this uses exactly the same sum of squares and degrees of freedom—for further detail see above). In Figure 2 we still show the main effect of genetic richness as well. However, as we found our alternative contrast coding more biologically meaningful and used it also for the separate SEMs, we focus the analysis on this contrast coding. Of course, also these contrasts are orthogonal and thus again independent of fitting sequence. In the revised version we tried to better explain the use of this alternative contrast coding.

Lines 420–428: “To determine how species and genetic diversity and their interaction affected tree functional diversity and trophic interactions, linear mixed-effects models (LMMs) were fitted with two types of contrast coding. In the first, we used the ordinary 2-way analysis of variance with interaction and in the second we replaced the genetic diversity main effect and the interaction with separate genetic diversity effects for species monocultures and the species mixture (Appendix 2—table 4). Note that as our design was orthogonal, fitting sequence did not matter in either of the codings. However, we focused our major analysis on the second type of coding to make it consistent with our hypotheses. Main effects of genetic diversity are presented in inset panels in Figure 2.”

Lines 120–121: “Using linear mixed-model analyses, we tested the effects of species diversity and genetic diversity within species on trophic interactions and community productivity.”

Figure 1c. Better name mother trees 1, 2, 3, and 4 to match the seed families naming.

Thank you for the suggestion. We have named the mother trees as “1, 2, 3, 4” to match with the names of seed families (see Figure 1c).

Figure 2. Why the direct link between species diversity and productivity is not shown? Please, clarify this in the figure or methods. In addition, the paths could be changed to more contrasting colors to facilitate the reading for color-blind people.

We removed the direct link between species diversity and productivity because it was a non-informative pathway and its removal reduced AICc of the path models. Now we have justified this in the Methods and the figure legend.

Lines 452–454: “We sequentially dropped non-informative pathways, if their removal reduced the AICc of the path models by more than 2 (Grace, 2006).”

Lines 794–797: “The direct effect of tree species diversity on tree community productivity was removed in the model because it was not significant (*P* > 0.5) and the removal reduced the AICc by more than 2 (ΔAICc = 3.269).”

Thank you for your suggestion for the color, now we have changed the colors of the paths and we used an online tool (https://coolors.co/ca5100-3f979a) to make sure the current colors are viewer-friendly.

References:

Abdala-Roberts, L., Mooney, K. A., Quijano-Medina, T., Campos-Navarrete, M. J., Gonzalez-Moreno, A., and Parra-Tabla, V. (2015). Comparison of tree genotypic diversity and species diversity effects on different guilds of insect herbivores. *Oikos, 124*(11), 1527-1535. https://doi.org/10.1111/oik.02033

Bardgett, R. D., Mommer, L., and De Vries, F. T. (2014). Going underground: root traits as drivers of ecosystem processes. *Trends in Ecology and Evolution, 29*(12), 692-699. https://doi.org/10.1016/j.tree.2014.10.006

Bates, D., Mächler, M., Bolker, B., and Walker, S. (2015). Fitting Linear Mixed-Effects Models Using lme4. *Journal of Statistical Software, 67*(1), 1 – 48. https://doi.org/10.18637/jss.v067.i01

Bruelheide, H., Nadrowski, K., Assmann, T., Bauhus, J., Both, S., Buscot, F., Chen, X. Y., Ding, B. Y., Durka, W., Erfmeier, A., Gutknecht, J. L. M., Guo, D. L., Guo, L. D., Hardtle, W., He, J. S., Klein, A. M., Kuhn, P., Liang, Y., Liu, X. J., Michalski, S., Niklaus, P. A., Pei, K. Q., Scherer-Lorenzen, M., Scholten, T., Schuldt, A., Seidler, G., Trogisch, S., von Oheimb, G., Welk, E., Wirth, C., Wubet, T., Yang, X. F., Yu, M. J., Zhang, S. R., Zhou, H. Z., Fischer, M., Ma, K. P., and Schmid, B. (2014). Designing forest biodiversity experiments: general considerations illustrated by a new large experiment in subtropical China. *Methods in Ecology and Evolution, 5*(1), 74-89. https://doi.org/Doi 10.1111/2041-210x.12126

Bustos-Segura, C., Poelman, E. H., Reichelt, M., Gershenzon, J., and Gols, R. (2017). Intraspecific chemical diversity among neighbouring plants correlates positively with plant size and herbivore load but negatively with herbivore damage. *Ecology Letters, 20*(1), 87-97. https://doi.org/https://doi.org/10.1111/ele.12713

Craven, D., Eisenhauer, N., Pearse, W. D., Hautier, Y., Isbell, F., Roscher, C., Bahn, M., Beierkuhnlein, C., Bönisch, G., Buchmann, N., Byun, C., Catford, J. A., Cerabolini, B. E. L., Cornelissen, J. H. C., Craine, J. M., De Luca, E., Ebeling, A., Griffin, J. N., Hector, A., Hines, J., Jentsch, A., Kattge, J., Kreyling, J., Lanta, V., Lemoine, N., Meyer, S. T., Minden, V., Onipchenko, V., Polley, H. W., Reich, P. B., van Ruijven, J., Schamp, B., Smith, M. D., Soudzilovskaia, N. A., Tilman, D., Weigelt, A., Wilsey, B., and Manning, P. (2018). Multiple facets of biodiversity drive the diversity–stability relationship. *Nature Ecology and Evolution, 2*(10), 1579-1587. https://doi.org/10.1038/s41559-018-0647-7

Grace, J. B. (2006). *Structural Equation Modeling and Natural Systems*. Cambridge University Press. https://doi.org/DOI: 10.1017/CBO9780511617799

Hillebrand, H., and Matthiessen, B. (2009). Biodiversity in a complex world: consolidation and progress in functional biodiversity research. *Ecology Letters, 12*(12), 1405-1419. https://doi.org/10.1111/j.1461-0248.2009.01388.x

Kahmen, A., Renker, C., Unsicker, S. B., and Buchmann, N. (2006). Niche complementarity for nitrogen: An explanation for the biodiversity and ecosystem functioning relationship? *Ecology, 87*(5), 1244-1255. https://doi.org/10.1890/0012-9658(2006)87[1244:Ncfnae]2.0.Co;2

Koricheva, J., and Hayes, D. (2018). The relative importance of plant intraspecific diversity in structuring arthropod communities: A meta-analysis. *Functional Ecology, 32*(7), 1704-1717. https://doi.org/10.1111/1365-2435.13062

Kuznetsova, A., Brockhoff, P. B., and Christensen, R. H. B. (2017). lmerTest Package: Tests in Linear Mixed Effects Models. *Journal of Statistical Software, 82*(13), 1-26. https://doi.org/10.18637/jss.v082.i13

Marquard, E., Weigelt, A., Temperton, V. M., Roscher, C., Schumacher, J., Buchmann, N., Fischer, M., Weisser, W. W., and Schmid, B. (2009). Plant species richness and functional composition drive overyielding in a six-year grassland experiment. *Ecology, 90*(12), 3290-3302.https://doi.org/10.1890/09-0069.1

Nabity, P. D., Zavala, J. A., and DeLucia, E. H. (2009). Indirect suppression of photosynthesis on individual leaves by arthropod herbivory. *Annals of Botany, 103*(4), 655-663. https://doi.org/10.1093/aob/mcn127

Niklaus, P. A., Baruffol, M., He, J.-S., Ma, K., and Schmid, B. (2017). Can niche plasticity promote biodiversity–productivity relationships through increased complementarity? *Ecology, 98*(4), 1104-1116.https://doi.org/10.1002/ecy.1748

Sapijanskas, J., Paquette, A., Potvin, C., Kunert, N., and Loreau, M. (2014). Tropical tree diversity enhances light capture through crown plasticity and spatial and temporal niche differences. Ecology, 95(9), 2479-2492. https://doi.org/10.1890/13-1366.1

Schuldt, A., Bruelheide, H., Härdtle, W., Assmann, T., Li, Y., Ma, K., von Oheimb, G., and Zhang, J. (2015). Early positive effects of tree species richness on herbivory in a large-scale forest biodiversity experiment influence tree growth. *Journal of Ecology, 103*(3), 563-571. https://doi.org/10.1111/1365-2745.12396

Semchenko, M., Leff, J. W., Lozano, Y. M., Saar, S., Davison, J., Wilkinson, A., Jackson, B. G., Pritchard, W. J., De Long, J. R., Oakley, S., Mason, K. E., Ostle, N. J., Baggs, E. M., Johnson, D., Fierer, N., and Bardgett, R. D. (2018). Fungal diversity regulates plant-soil feedbacks in temperate grassland. Science Advances, 4(11), eaau4578. https://doi.org/10.1126/sciadv.aau4578

van Moorsel, S. J., Hahl, T., Wagg, C., De Deyn, G. B., Flynn, D. F. B., Zuppinger-Dingley, D., and Schmid, B. (2018). Community evolution increases plant productivity at low diversity. Ecology Letters, 21(1), 128-137. https://doi.org/10.1111/ele.12879

van Moorsel, S. J., Schmid, M. W., Wagemaker, N. C. A. M., van Gurp, T., Schmid, B., and Vergeer, P. (2019). Evidence for rapid evolution in a grassland biodiversity experiment. Molecular Ecology, 28(17), 4097-4117.https://doi.org/10.1111/mec.15191

Wetzel, W. C., Kharouba, H. M., Robinson, M., Holyoak, M., and Karban, R. (2016). Variability in plant nutrients reduces insect herbivore performance. Nature, 539(7629), 425-427. https://doi.org/10.1038/nature20140

Williams, L. J., Paquette, A., Cavender-Bares, J., Messier, C., and Reich, P. B. (2017). Spatial complementarity in tree crowns explains overyielding in species mixtures. Nature Ecology and Evolution, 1(4), 0063. https://doi.org/10.1038/s41559-016-0063

Zuppinger-Dingley, D., Schmid, B., Petermann, J. S., Yadav, V., De Deyn, G. B., and Flynn, D. F. B. (2014). Selection for niche differentiation in plant communities increases biodiversity effects. Nature, 515(7525), 108-111. https://doi.org/10.1038/nature13869

Züst, T., and Agrawal, A. A. (201). Trade-Offs Between Plant Growth and Defense Against Insect Herbivory: An Emerging Mechanistic Synthesis. Annual Review of Plant Biology, 68(1), 513-534. https://doi.org/10.1146/annurev-arplant-042916-040856

Zvereva, E. L., Zverev, V., and Kozlov, M. V. (2012). Little strokes fell great oaks: minor but chronic herbivory substantially reduces birch growth. Oikos, 121(12), 2036-2043.https://doi.org/10.1111/j.1600-0706.2012.20688.x

[Editors’ note: further revisions were suggested prior to acceptance, as described below.]

The manuscript has been improved but there are some remaining issues that need to be addressed, as outlined below:Both reviewers find that using species family means for trait values is problematic (for example, the assigning of zero values in Rao's Q calculations seems puzzling). While you have made a case for continuing to use family means in your R1 response, we find that a new opportunity must be given to sustain using these data, and explain possible artifacts when using zero values in, for example, your SEM paths. Possible alternatives (such as a SEM without the tests/paths between genotypic diversity and FDis due to colinearity issues when using family means) should be discussed (in an Appendix?), and the Discussion should also be extended, to make it clear this is a shortcoming of your analysis.

Thanks for the positive comments on the revision. We agree that the zero values of functional diversity in genetic monocultures of single species (1.1 diversity level) can be criticized, but we also believe the mean method has the advantages we mention in cases where it is not zero by definition. We now discuss the caveats and increase focus on the new method using individual trait values to calculate functional diversity as:

“We found that tree species diversity increased tree productivity via increased tree functional diversity, reduced soil fungal diversity and marginally reduced herbivory. The effects of tree genetic diversity on productivity via functional diversity and soil fungal diversity were negative in monocultures but positive in the mixture of the four tree species tested.” (lines 41-45)

“The results obtained with functional traits calculated from measurements on individual trees showed weaker effects of genetic diversity on functional diversity (path coefficient = 0.193 vs 0.883) but did not change the significance and direction of the effects of genetic diversity on productivity via functional diversity in species monoculture. Additionally, the effects of functional diversity on tree productivity in species mixtures were positive when using functional diversity calculated from measurements on individual trees but were non-significant when using functional diversity calculated from seed-family means (Figure 5, Figure 5—figure supplement 1).” (lines 178-185)

“Using functional diversity calculated from measurements on individual trees did not change the effects of genetic diversity via trophic feedbacks, except that the effects of herbivory on productivity became non-significant from marginally significant.” (lines 188-190)

“Regarding our first hypothesis, we found that tree species diversity and genetic diversity can increase tree community productivity via increased functional diversity and trophic feedbacks as predicted. This suggests complementary resource-use and biotic niches, respectively, as mechanisms underpinning the biodiversity effects (Turnbull et al., 2016). Nevertheless, compared with effects of species diversity, effects of genetic diversity on tree community productivity through functional diversity were weaker, whereas effects of genetic diversity on trophic interactions were strong (see Figure 4, Figure 4—figure supplement 1), indicating that mechanisms underpinning effects of genetic diversity may in part differ from those underpinning effects of species diversity, as we will discuss below. Regarding our second hypothesis, we found that the effects of tree genetic diversity on productivity via functional diversity and soil fungal diversity were negative in tree species monocultures but positive in the species mixture, which differed from our predictions. In the following, we discuss these results in more detail.” (lines 200-212)

“At the same time, the results indicate that the seed-family means method may bring an artifact to the effect of genetic diversity on functional diversity because of the zero value of functional diversity in genetic monocultures of single species (1.1 communities). However, excluding the path between genetic diversity and functional diversity did not affect remaining paths, indicating that the partly artificial relationship between genetic diversity and functional diversity did not distort the path model in general (Figure 4, Appendix 3—figure 1).” (lines 244-250)

“In contrast of our second hypothesis, we found that the effects of genetic diversity via functional diversity and soil fungal diversity were negative in species monocultures but not significant via functional diversity and positive via soil fungal diversity in the species mixture (Figure 5). We found genetic diversity had positive effects on tree functional diversity and soil fungal diversity in species monocultures but negative effects in the species mixture, which supports the trade-offs between genetic and species diversity discussed in the previous section.” (lines 269-274)

“The two methods of calculating functional diversity either from seed-family means or from trait values of individual trees yielded different results regarding the indirect effects of genetic diversity on tree productivity via functional diversity. The method based on seed-family means has the advantage to be less circular whereas the method based on trait values of individuals has the advantage of producing functional diversity values >0 also for genetic monocultures of single species (1.1 communities; see Methods). The weaker indirect effects of genetic diversity on tree productivity via functional diversity in the method using trait values of individuals suggests that the zero value of functional diversity in 1.1 communities in the method using seed-family means may lead to an overestimation of these indirect effects of genetic diversity in species monocultures. Nevertheless, the method using seed-family means is still useful for species monocultures with multiple seed families and for species mixtures.” (lines 283-293)

We have added SEMs without the paths between genetic diversity and functional diversity as Appendix 3 (Appendix 3—figure 1, Appendix 3—figure 2). Fortunately, the remaining path coefficients were not distorted by functional diversity and changed only little if the path from genetic diversity to functional diversity was omitted. We also added multi-group SEMs in which the path between genetic diversity and functional diversity calculated from seed-family means was excluded, the coefficients of the remaining paths also were not changed (Appendix 3—figure 2). We have included these analyses in the new version (lines 170-172, lines 190-193, lines 237 -250, lines 479-481 and lines 487-488).

Reviewer #1 (Recommendations for the authors):I have reviewed this revised version of the manuscript after already reviewing the original version (reviewer 1). In my opinion, the manuscript significantly improved and is much clear now. In particular, the detailed comments of reviewer 2 must have helped to clarify the methods, which in turn helped to better understand the results and discussion. I am overall happy with the analyses and also appreciate very much the inclusion of the results with traits measured at the individual level. I understand the reason of the authors with respect to circularity, but also maintain my concerns with using species family means, that do not only eliminate the potential responses of the genotypes to the treatments (which was the goal of the authors with using species family means) but also the potential effects of the genotypes. It is basically impossible to eliminate the response of a trait without also eliminating its effect. And in this study, I think this is particularly tricky because the results turn out to be significantly different when using species family means versus individual values, with important implications for the objectives of this study. And here I am in serious doubt whether the way the authors decided to go forward is the most appropriate one. I explain this based on the key result affected:

Thanks for the constructive comments on the revision. We agree that the seed-family means method is not a perfect way to show the functional variation among genotypes, especially for the 1.1 communities. However, because this is the common method used in the literature and because it has some advantages when it is not zero by definition, we would still like to use it, but now with the increased focus on the new method using trait values of individual trees to calculate functional diversity. We have extended our discussion and adjusted the interpretation by comparing the results obtained with the two methods (see the above text pieces taken from the main text in the response to the editor’s comments).

L263ff: I find the results of negative genotype diversity effects on productivity in species monocultures puzzling. And it, in fact, only appears in the SEM and only when using the species family means of trait values but not the measured trait values in each subplot. I can't really make a conclusion out of this, but it seems weird to me, and I really wonder to what extent this is due to the assumed 0 functional diversity of genotype monocultures (1.1 communities). I can't help but think this is an artifact. Actually, the SEM with the functional traits measured on individual trees matches much better the results observed in the binary analyses, where genotype diversity doesn't have an impact on functional diversity of species monocultures nor community productivity of species monocultures (Figure 2 and its supplement 1). It seems rather that increased functional diversity in species monocultures goes along with reduced productivity (Figure 3), which is, however independent of tree genetic diversity (as there is no clear relationship between tree genetic diversity and functional diversity of species monocultures as stated above – when individual trait values are used). So, it seems to me that the negative effect of tree genetic diversity on productivity in species monocultures, as claimed by the authors, is not a genetic diversity effect but a functional diversity effect independent of genetic diversity. That would at least be my interpretation of the results.

In binary analyses, we did not find the overall negative effects of genetic diversity on productivity, neither in monoculture nor mixture (Figure 2a). However, we found genetic diversity had overall positive effects on functional diversity calculated from seed-family means and showed different effects between species monocultures and mixtures (Figure 2b), in part possibly because of 1.1 communities forced to have zero functional diversity. We also found that functional diversity had significant effects on tree productivity by both when calculated from seed-family means or from trait values of individual trees (Figure 3a, Figure 3a—supplementary 1). According to these binary results, we hypothesized that genetic diversity has the potential to affect productivity via functional diversity and the effects might differ between species monocultures and mixtures. So, we carried out multi-group structural equation models to test this hypothesis (Figure 5, Figure 5—supplementary 1). To show the effects of the zero value in 1.1 communities, we also calculated the structural equation model based on the functional diversity calculated by individual values. We found that the negative effects of genetic diversity on productivity in species monoculture were consistent in both methods (Figure 5 vs Figure 5—supplementary 1). According to these results, we concluded that although we did not find genetic diversity had significant effects on productivity in binary analysis, we found that it negatively affected productivity in species monocultures when we considered the indirect effect via functional diversity on productivity. Now we extended the discussion and adjusted the interpretation to make the results clearer and state the potential effects of zero values in 1.1 communities (see the above text pieces taken from the main text in the response to the editor’s comments).

Reviewer #2 (Recommendations for the authors):I thank the authors for thoroughly answering all the reviewers' questions and comments. I think the present version clarifies all the points raised and shows more clarity in the design and discussion.I am glad to read this revised version and congratulate the authors for such an impressive and interesting study that represent a step forward in our understanding of the effects of plant diversity.

Thanks for the positive comments on the revised manuscript.